# Automatic Curriculum Expert Iteration for Reliable LLM Reasoning

**Zirui Zhao**[1]* **Hanze Dong**[2]† **Amrita Saha**[2] **Caiming Xiong**[2] **Doyen Sahoo**[2]
[1]National University of Singapore    [2]Salesforce AI Research

## Abstract

Hallucinations (i.e., generating plausible but inaccurate content) and laziness (i.e. excessive refusals or defaulting to "I don't know") persist as major challenges in LLM reasoning. Current efforts to reduce hallucinations primarily focus on factual errors in knowledge-grounded tasks, often neglecting hallucinations related to faulty reasoning. Meanwhile, some approaches render LLMs overly conservative, limiting their problem-solving capabilities. To mitigate hallucination and laziness in reasoning tasks, we propose **Auto**matic **C**urriculum **E**xpert **I**teration (Auto-CEI) to enhance LLM reasoning and align responses to the model's capabilities—assertively answering within its limits and declining when tasks exceed them. In our method, Expert Iteration explores the reasoning trajectories near the LLM policy, guiding incorrect paths back on track to reduce compounding errors and improve robustness; it also promotes appropriate "I don't know" responses after sufficient reasoning attempts. The curriculum automatically adjusts rewards, incentivizing extended reasoning before acknowledging incapability, thereby pushing the limits of LLM reasoning and aligning its behaviour with these limits. We compare Auto-CEI with various SOTA baselines across logical reasoning, mathematics, and planning tasks, where Auto-CEI achieves superior alignment by effectively balancing assertiveness and conservativeness. [1]

## 1 Introduction

Hallucination is a long-standing issue in large language model (LLM) research, which refers to the phenomenon where LLM-generated content appears plausible but is actually nonsensical or inaccurate, often misleading humans by seeming deceptively credible (Ji et al., 2023). It is more evident when solving complex reasoning problems beyond their capability, in which LLM tends to fake evidence or logic to answer the questions assertively. The reason for LLM's hallucinations, overall, is a misalignment of its real capability and its behaviours: LLM should not behave overly assertively or confidently when facing unfamiliar (Kang et al., 2024a) and difficult problems (Dziri et al., 2024). Instead, a preferred behaviour of LLMs is to acknowledge their limitations by responding with "I don't know." However, we must also prevent the LLM from defaulting to degenerate responses like "I don't know," especially when the question is within its capability. A reliable LLM should strike a balance between maximizing performance and avoiding hallucination.

Various methods have been proposed to mitigate LLM hallucinations. However, most studies have focused on *factual hallucinations*, i.e., fabricating non-existent evidence, while often neglecting *reasoning hallucinations* (Creswell et al., 2022). Reasoning tasks require deriving conclusions from evidence using rules, often involving multiple steps. *Reasoning hallucination* refers to the phenomenon where LLMs apply invalid rules or misinterpret the conclusion, leading to an incorrect final result even without factual hallucination in the evidence. Dziri et al. (2024) found that LLMs exhibit a probability of error at each reasoning step. As these errors compound, the likelihood of incorrect final answers increases exponentially as the reasoning chain grows longer. In addition, most techniques mitigate factual hallucinations using external knowledge or conduct post-training to do "patching". These techniques can reasonably mitigate factual errors, but hallucinations caused

---

*Work was done when Zirui Zhao interned at Salesforce AI Research, Singapore.

†Correspondence to Hanze Dong (hanze.dong@salesforce.com).

[1]The code is available at `https://github.com/SalesforceAIResearch/Auto-CEI`.

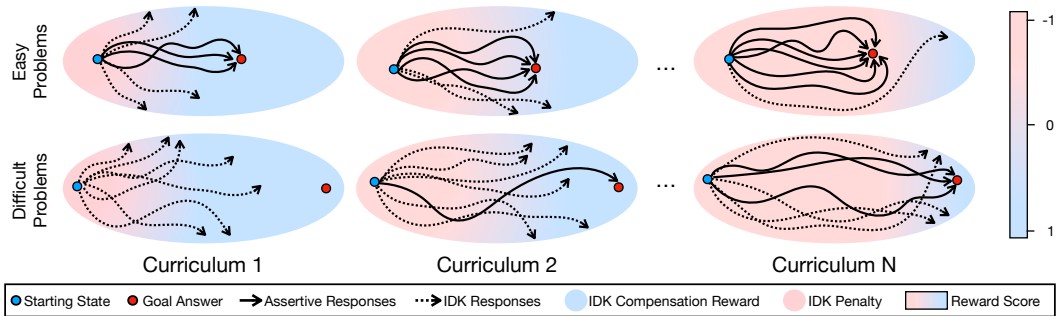

Figure 1: Demonstration of the key idea of AUTO-CEI. AUTO-CEI automatically designs a sequence of curricula and uses Expert Iteration to train LLM in each curriculum. Expert Iteration explores the reasoning trajectories (i.e., Chain of thought) and learns to get incorrect paths back on track to reduce compounding errors. Using the reward (denoted by blue and red colour) produced by the curriculum, Expert Iteration also samples appropriate "I don't know" responses (IDK responses) after sufficient reasoning attempts (in the blue region). To mitigate laziness, it avoids the short IDK responses (in the red region). The curriculum automatically adjusts rewards for the reasoning length, incentivizing extended reasoning before acknowledging incapability, thereby pushing the limits of LLM reasoning and aligning its assertive and conservative behaviours with these limits.

by faulty reasoning are far more challenging due to the inevitable compounding error of the transformer. As such, most of these techniques cannot improve LLM's inherent reasoning capability, and simply using patching techniques alone can even make LLMs behave lazily.

Can LLM learn *short reasoning* or *self-correction* to reduce compounding errors and reasoning hallucinations? Kang et al. (2024b) showed that learning the optimal shortest solution for difficult problems (e.g., NP-complete problems) has very high sample complexity. Thus, directly learning short solutions for all complex problems is unrealistic. Havrilla et al. (2024) showed that reinforcement learning (RL) can help improve LLM's performance in various reasoning tasks. RL methods, such as Expert Iteration (Anthony et al., 2017) and PPO (Schulman et al., 2017), explore various LLM reasoning trajectories (i.e., Chain of Thought (Wei et al., 2022)) and use reward function to guide the incorrect trajectories back on track, therefore improving the robustness and reducing the effect of compounding errors. However, compounding errors are inherently impossible to eliminate. Even though RL reduces step-wise errors, the probability of error still increases exponentially with the length of reasoning. As a result, it is necessary for LLMs to acknowledge their limitations, particularly when faced with difficult problems beyond their capability. This raises a key question:

> *How can we assess the limits of LLMs' reasoning abilities and adjust their response behaviour appropriately to align with these limits?*

To address this, we propose Automatic Curriculum Expert Iteration (AUTO-CEI), which simultaneously enhances LLM reasoning and aligns its behaviour to ensure precise answers while acknowledging its limitations. AUTO-CEI assumes that the number of reasoning steps required to reach a correct answer provides a reasonable estimate of both the problem difficulty and the limits of LLM reasoning. This assumption is grounded in computational theory, as each reasoning problem has its underlying computational complexity, and each reasoning step corresponds to an elementary computing operation. Learning the reasoning steps (precondition/effect of reasoning rules) has similarly low sample complexity (Kang et al., 2024b). Despite the existence of concise optimal solutions, difficult problems (e.g., NP-complete problems) require extended reasoning to find those solutions. This assumption helps align LLM's behaviours: easier problems require fewer reasoning steps and are less prone to compounding errors, justifying greater assertiveness from LLMs. Complex problems needing more steps and suffering more compounding errors require more conservativeness.

The key idea of AUTO-CEI is shown in Figure 1: using the length of reasoning steps to measure difficulty, AUTO-CEI designs an automatic curriculum for expert iteration that rewards correct reasoning, compensates for "I don't know" acknowledgements after a sufficient number of attempts, and penalizes both overly conservative and assertive wrong responses. The expert iteration adjusts the LLM's responses by resampling based on these rewards. The curriculum automatically updates the compensation reward to encourage more reasoning attempts before saying "I don't know" over time. It gradually adjusts the reward function to optimise an objective function, balancing the overall precision and the proportion of saying "I don't know" to control hallucination and avoid laziness.

As such, it gradually pushes the limits to maximise the potential of LLM reasoning and aligns its behaviours with these limits. AUTO-CEI is effective across various reasoning tasks such as logical, mathematical, and planning problems, balancing reliability and conservativeness.

**Contributions.** In summary, AUTO-CEI automatically estimates the boundary of its reasoning capacity, thus achieving a reasonable alignment to maximise capacity and control behaviours. It learns to reliably solve the problems within its boundary as much as possible; it also knows to acknowledge "I don't know" when the problem is beyond its limit. We carried out comprehensive experiments in various reasoning benchmarks, and AUTO-CEI significantly outperforms the concurrent baseline methods, boosting precision by 10-24% while maintaining a relatively low refusal rate of 18-36% across diverse reasoning tasks in planning, logical and math reasoning.

## 2 RELATED WORK

**Hallucinations and laziness of LLM.** It is widely acknowledged that LLMs often produce hallucinated responses, including fabricated facts and misleading logic (Radhakrishnan et al., 2023; Ji et al., 2023). Most works focus on reducing hallucinations by ensuring the factual accuracy of generated content, often using retrieval-based methods that enhance LLMs with external knowledge (Shuster et al., 2021). This methodology has been effective in improving the reliability of LLMs in domains that require high factual accuracy, such as knowledge-based question answering (Lewis et al., 2020). Particularly, Retrieval-Augmented Generation (RAG) reduces hallucinations by fetching relevant documents from external sources during inference, integrating factual information to enhance accuracy and relevance (Izacard et al., 2023). These methods ensure that the LLMs' responses align more closely with verified data (Komeili et al., 2022; Gururangan et al., 2021). Moreover, several studies have explored hybrid approaches that combine retrieval with fact-checking modules or domain-specific fine-tuning to improve factual accuracy (Lee et al., 2022). Retrieval-based strategies show strong potential in mitigating hallucinations, outperforming purely generative models that rely only on knowledge from training (Borgeaud et al., 2022; Petroni et al., 2020). However, hallucinations are not limited to factual inaccuracies; they can also extend to faulty or incomplete reasoning, a significant concern for multi-step reasoning-based tasks (Creswell et al., 2022). Moreover, LLMs often exhibit what can be described as "laziness," which refers to the tendency of the model to reject or avoid generating correct but complex answers in favour of simpler, superficial, or incomplete responses (Bubeck et al., 2023; Bang et al., 2023). This phenomenon has been noted in tasks requiring step-by-step logical reasoning, where LLMs tend to skip intermediate steps or provide too general answers, rather than addressing the nuanced complexity of the problem at hand (Rae et al., 2021).

**Reinforcement learning for LLM reasoning.** Reinforcement learning has been widely used in the alignment stage of LLM training to enhance their capabilities. The research in this area can be broadly divided into two aspects: (1) RL algorithms; (2) reward construction. Conventional advanced deep RL algorithms, such as Proximal Policy Optimization (PPO) (Schulman et al., 2017), have been extensively applied in the training process of modern LLMs, helping to improve both format and reasoning abilities (Achiam et al., 2023; Team et al., 2023; Dubey et al., 2024). However, due to the introduction of the value network, the computational overhead of these algorithms is extremely high, often making them unaffordable for academic settings where resources are limited. More recently, expert iteration (also known as rejection sampling or reward-ranked fine-tuning) (Dong et al., 2023; Gulcehre et al., 2023; Zelikman et al., 2022; Hoffman et al., 2024), preference-based RL (Rafailov et al., 2024; Xiong et al., 2024a), and REINFORCE-type RL (Williams, 1992; Shao et al., 2024; Li et al., 2023; Ahmadian et al., 2024) have emerged as more efficient alternatives for the alignment stage, especially when operating within a fixed budget. These methods have demonstrated significant improvements over supervised fine-tuning in reasoning tasks (Aksitov et al., 2023; Gou et al., 2023; Havrilla et al., 2024; Shao et al., 2024; Dong et al., 2024; Trung et al., 2024; Xiong et al., 2024b). Additionally, Lightman et al. (2023) introduced process supervision to refine the reward signal for each reasoning step. Compared to traditional outcome-based supervision, process supervision has been shown to yield substantial gains (Lightman et al., 2023; Wang et al., 2024a; Shao et al., 2024). Despite the impressive progress in this area, most existing works focus on maximizing the upper bound of reasoning ability. In contrast, our algorithm aims to develop a more balanced model incorporating self-awareness into the reasoning procedure.

## 3 PRELIMINARIES

### 3.1 FORMULATION: REASONING PROBLEMS

We address reasoning problems in general, such as logical reasoning, mathematical reasoning, and planning. Reasoning involves deriving unknown facts from provided evidence and reasoning rules. In logical reasoning, evidence is represented as clauses with assigned boolean values, and applying reasoning rules determines the boolean values of unknown clauses. In arithmetic reasoning, evidence consists of clauses linked to scalar values, and reasoning requires both logical deduction and arithmetic operations. In planning problems, the goal is to identify a sequence of actions that move an initial state to a target state by applying actions based on their preconditions and effects.

Unlike methods that convert problems into Boolean satisfiability (SAT) and solve them with SAT solvers, the LLM reasoning approach, such as Chain of Thought (CoT) (Wei et al., 2022), follows a sequential decision-making strategy. To capture reasoning problems, we rely on planning concepts that consist of four main elements: state space $S$, action space $A$, transition function $T$, and goal function $G$. The state space $S$ includes all possible configurations, such as the boolean or scalar values assigned to the given variables or clauses, as well as unknown clauses. Actions, defined in the action space $A$, represent rules or equations that can be applied to these states. The action has its preconditions and effects. A state should satisfy the corresponding precondition to execute an action, and then the effects will be applied to the state. The precondition and effect are encoded in the transition function. When an action $A$ is executed in a state $S$, the transition function $T(s, a)$ determines the resulting state $s'$, e.g., new values would be assigned to (unknown) clauses or variables. The goal function $G$ checks whether the current state matches the desired outcome.

### 3.2 BACKGROUNDS

**Expert Iteration.** Expert iteration (EI) (Anthony et al., 2017) is an iterative process in which experts are built upon a base policy, the base policy is refined by learning from the expert, and this cycle repeats with newly derived base policies. EI is effective in improving the quality of generated responses in LLM literature (Dong et al., 2023; Gulcehre et al., 2023). Havrilla et al. (2024) discussed the usage of EI in LLM reasoning problems. They used LLM sampling to generate responses, and preferred responses were selected by a ground-truth or reward model to build the expert. They then used LLM to imitate the expert via SFT. EI explores the neighbouring area of LLM policy in the token space, which helps LLM derive back to the correct reasoning trajectories if there are errors or find suboptimal trajectories. As a result, in their experiment, EI improved the LLM reasoning over time, which performed similarly or even better than the PPO (Schulman et al., 2017).

**Curriculum Reinforcement Learning.** Curriculum Reinforcement Learning (Narvekar et al., 2020) focuses on sequencing tasks or data samples into a structured curriculum to solve complex problems that are difficult to learn from scratch. A curriculum can be considered an ordering over experience samples at the most basic level. The underlying assumption is that learning from elementary samples or tasks can be transferred and help learn more complex tasks. In this paper, we design the curriculum by reshaping the reward to gradually guide the LLM in conducting more reasoning attempts to solve the problem before acknowledging its limits. Its attempts could be used to discover suboptimal trajectories that mitigate its compounding errors in easy problems and also transferred to increase the chances of solving more complex problems.

## 4 AUTOMATIC CURRICULUM EXPERT ITERATION

AUTO-CEI aims to estimate and push the limits of LLM reasoning concerning the number of reasoning steps, maximising its capacity while aligning its behaviours accordingly. We assume that, for a broad class of reasoning problems, the number of reasoning steps (in the chain of thoughts) required to reach the correct answer estimates the problem's difficulty and the limits of LLM reasoning. We justify the assumption by computational theory. The reasoning problem has its underlying computational complexity, where each of the reasoning steps can be treated as an elementary computing operation. Learning to apply the preconditions and effects of those elementary reasoning actions/rules has low sample complexity (Kang et al., 2024b). Although problems may have concise optimal solutions, given the inherent computational complexity (e.g., NP-complete), finding that

solution still needs long reasoning attempts. AUTO-CEI estimates the reasoning limits in terms of the number of reasoning steps until the LLM can respond reliably, solving a maximum number of problems. But beyond this point, further continuing the same assertive behaviour of the LLM policy leads to a significant increase in mistakes, whereas being overly conservative within this range limits reasoning capacities due to the high refusal rate. Thus, it estimates the LLM's capability's boundary when fine-tuned for multi-step reasoning tasks.

AUTO-CEI uses the average number of the reasoning steps of the initial (after supervised finetuning, SFT) policy to determine the curriculum, i.e., the reward function. It starts from optimising an initialised policy that is reasonably conservative with a certain distribution to say, "I don't know." It then runs into the Expert Iteration process, in which the LLM policy will sample many reasoning trajectories and receive various rewards. We build up the expert by resampling the trajectories according to the reward and removing the assertively wrong responses. After convergence, AUTO-CEI updates the reward function according to the performance, in a way to encourage the LLM to produce more reasoning steps before saying "I don't know." The curriculum eventually tries to optimise an objective function, which balances assertiveness and conservativeness and stops the curriculum if the objective function reaches the optimal value.

## 4.1 INITIALISATION

We adopt R-Tuning (Zhang et al., 2023), one of our baseline methods, as the initialization strategy. R-Tuning produces a good starting point with a reasonable proportion of refusal behaviours. It also makes LLM lazier and has a lot of room for optimisation. We first use SFT to train a language model given the training dataset $D$ consisting of $n$ question-answer

---

**Algorithm 1** INITIALISATION$(D, \pi)$

1: $\pi_{\text{sft}} \leftarrow \text{SFT}(\pi, D)$
2: $D' \leftarrow \{(x', y') \mid y' \sim \pi_{\text{sft}}(x'), x' \in D\}$
3: Add IDK to all wrong $y'$ in $D'$ to get $D_{\text{init}}$
4: $\pi_0 \leftarrow \text{SFT}(\pi, D_{\text{init}})$
5: **return** $\pi_0$

---

pairs. Each answer uses the chain of thought. After SFT, we will let the language model answer the questions in the same dataset $D$ using random sampling. As we use random sampling, there will be a certain number of correct answers, and the others will be wrong. Thus, we split the questions and the new answers generated by LLM into two datasets, $D_1$ and $D_2$, where $D_1$ collects correct answers and $D_2$ consists of wrong answers. For $D_2$, in each of the wrong answers, we add an expression at the end to acknowledge the limitations and abstain from answering the questions, such as "*Sorry, I am not sure if I answer the question correctly. There might be mistakes in my answer.*" Thus, we have collected a new Refusal-Aware dataset $D_2$ for our finetuning.

After collecting $D_1$ and $D_2$, we concatenate both datasets to form $D_{\text{init}}$ and use SFT to fine-tune the LLM again. We also examined the SFT result to ensure it had enough sample points for assertive and refusal answers. We set a threshold of 25% for the distribution of refusal behaviours in the validation set, as we need to have enough variety of acknowledge "I don't know" for AUTO-CEI. If the model has a very low distribution of acknowledge "I don't know", we then repeat the previous process of collecting $D_1$ and $D_2$ again, where the refusal answers will also be collected in $D_1$ and use the new dataset for a new SFT training. We observed that, empirically, the distribution of refusal answers would increase over the steps. Thus, after the initialisation, we will have a good initial policy with enough base knowledge to answer questions and a distribution of acknowledging "I don't know."

## 4.2 EXPERT ITERATION

Given the initial policy, we run the Expert Iteration to improve the policy over time.

We build the expert using the reward function and resampling. The reward function $R(x, y)$ takes the question $x$ and an answer $y$ as input and outputs a scalar. $R(x, y)$ is designed to give a large positive reward for correct answers, a small compensatory reward for refusal answers after long reasoning trajectories, a small negative penalty for refusal answers after very short reasoning trajectories, and a large negative penalty for assertive wrong answers. The re-

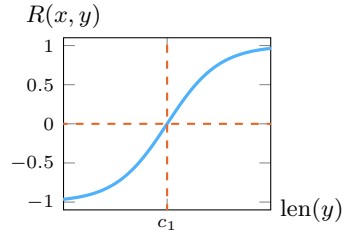

Figure 2: The shape of $R(x, y)$ for $y$ is a refusal answer.

ward is designed as follows:

$$R(x, y) = \begin{cases} 1 & \text{for } y \text{ correctly answers } x \text{ ;} \\ \frac{1-\exp(-c_2(\text{len}(y)-c_1))}{1+\exp(-c_2(\text{len}(y)-c_1))} & \text{for } y \text{ is a refusal answer ;} \\ -1 & \text{for } y \text{ wrongly answers } x \text{ .} \end{cases} \quad (1)$$

In this function, $c_1$ and $c_2$ are two hyperparameters determined by the distribution of answers from the initial LLM. If $y$ is a refusal answer and the number of steps for the reasoning trajectory $\text{len}(y)$ is longer than $c_1$, then it will receive a small positive compensation reward; otherwise, it turns into a small negative penalty. The shape of this function is shown in Figure 2. $c_1$ is initialised by the mean value of reasoning steps produced by the initial LLM policy in the validation set. $c_2$ is computed by solving $\frac{1-\exp(-c_2 \cdot 2\sigma)}{1+\exp(-c_2 \cdot 2\sigma)} = 0.9$, meaning that if the number of reasoning steps has reached $c_1 + 2\sigma$ ($\sigma$ is the standard deviation of the reasoning steps by initial policy), then the reward should be higher than 0.9. It means the answer is longer than roughly 97% of the reasoning trajectories in the validation set, assuming the distribution of reasoning steps is a normal distribution.

The reward function helps us build the expert, enabling us to improve LLM reasoning performance over time. We resample the responses according to the rewards to build the expert. We first use LLM to generate $K$ samples $\{y_i\}_{i=1}^K$ for each of the questions in the training dataset $D$. Then, we use the reward function to get a reward value of each sample $r_1, r_2, \ldots, r_K$. Over that reward, we employ the temperature-scaled softmax function to construct a new distribution: $p'(y_i) = \exp(r_i/\tau) / \sum_j \exp(r_j/\tau)$ and subsequently use $p'(y_i)$ to conduct resampling and generate $N$ new samples $\{y_i'\}_{i=1}^K$. $\tau$ is the temperature of resampling, the same as

---

**Algorithm 2** EXPERTITER($D, \pi_{\text{ei}}, R, D_{\text{val}}, \pi$)

1: **while** $\pi_{\text{ei}}$ doesn't converge in $D_{\text{val}}$ **do**
2:    **for** $x \in D$ **do**
3:       $Y_x \leftarrow \{y_i | y_i \sim \pi_{\text{ei}}(x)\}_{i=1}^K$
4:       $R_x \leftarrow \{r_i | r_i = R(x, y_i), y_i \in Y_x\}$
5:       Resamples $Y_x$ according to $R_x$ to form new responses set $Y_x'$
6:    **end for**
7:    $D_{\text{new}} \leftarrow \{(x, y) \mid y \in Y_x', x \in D\}$
8:    $\pi_{\text{ei}} \leftarrow \text{SFT}(\pi, D_{\text{new}})$
9: **end while**
10: **return** $\pi'$

---

the overall accuracy of the initial SFT model finetuned by $D$. If the SFT model has high overall accuracy, we will encourage LLM to explore more randomly; otherwise, the responses with higher rewards will be sampled more densely. $\tau$ will be capped in a range $[0.4, 0.7]$ to avoid extreme cases. Thus, the distribution is reshaped to encourage more instances that produce correct answers and produce more reasoning steps before saying "I don't know." Since resampling also kills instances that waste some of the exploration data, we concatenate the $\{y_i\}_{i=1}^K$ and $\{y_i'\}_{i=1}^K$ together to form the new training dataset. Lastly, we removed the answers that had $-1$ rewards.

## 4.3 CURRICULUM UPDATE

After EXPERTITER converges, the curriculum updates the reward function to optimise (i.e. maximise) an objective function:

$$f = (1 - \lambda)P_{\text{Pre}} + \lambda(1 - P_{\text{IDK}}), \quad (2)$$

where $P_{\text{Pre}}$ denotes the precision, i.e., the correctness of answers that are not refusal answers, and $P_{\text{IDK}}$ denotes the proportion of refusal answers. This function $f$ tries to find a good balance between

---

**Algorithm 3** AUTO-CEI($D, \pi, R, f, D_{\text{val}}$)

1: $\pi_{\text{ei}} \leftarrow \text{INITIALISATION}(D, \pi)$
2: **while** Optimal $f$ is not found **do**
3:    Update $c_1$ in reward function $R$   ▷ Eq 1
4:    $\pi_{\text{ei}} \leftarrow$ EXPERTITER($D, \pi_{\text{ei}}, R, D_{\text{val}}, \pi$)
5:    Measure $f$ by $\pi_{\text{ei}}$ and $D_{\text{val}}$
6: **end while**
7: **return** $\pi_{\text{ei}}$

---

avoiding both hallucination (low $P_{\text{Pre}}$ and low $P_{\text{IDK}}$) and laziness (high $P_{\text{Pre}}$ but high $P_{\text{IDK}}$) when finetuning the LLM. $\lambda \in [0, 1]$ is a hyperparameter to control the tradeoff between hallucination and laziness; a higher value of $\lambda$ will lead to more assertive behaviour (i.e. lower IDK rate) while smaller $\lambda$ would make the LLM policy behave more conservatively. We empirically show that setting $\lambda$ is fairly intuitive; in most task scenarios, setting $\lambda$ to a reasonably small value ($\approx 0.2$) achieves the desired effect for the LLM policy. Over the EI iterations, the curriculum will update the value of $c_1$ in Equation 1, i.e., try to push $c_1$ to encourage more exploration before saying "I don't know". Thus, with the new reward function, we repeat the Expert Iterations and make it converge again.

Assuming that the function of $f$ does not have local optimal point wrt $c_1$, the curriculum uses local search (hill climbing) (Russell & Norvig, 2016) to search $c_1$ and conduct Expert Iterations until it

finds the highest $f$. The hill-climbing algorithm searches its neighbouring value of $c_1 \pm d$ via a step size $d$, and if $f$ at the new step is higher, then it updates $c_1$ to the new value. It will stop the search if its neighbouring $c_1$ has a smaller value than its current $f$. The domain of $c_1$ is $[\mu - 2\sigma, \mu + 2\sigma]$ where $\mu$ denotes the average length of the reasoning steps produced by the initial LLM policy. $d$ is defined by $\min\{0.5, 4\sigma/10\}$, where $\sigma$ denotes the standard deviation of the reasoning length produced by the initial LLM policy. We also do not want to make the step too large, thus capping it by a threshold value of $0.5$, an empirically selected value.

Overall, the curriculum update, together with initialisation and Expert Iteration of AUTO-CEI, are in the pseudo-code listed in Algorithm 3.

## 5 EXPERIMENT

### 5.1 EXPERIMENTAL SETUP

**Benchmarks.** To demonstrate the effectiveness of AUTO-CEI in reasoning tasks, we select BoardgameQA (Kazemi et al., 2024), MATH (Hendrycks et al., 2021), and Blocksworld (Valmeekam et al., 2023) as benchmarks, spanning from logical and mathematical reasoning to planning. They have various domains and complexities. We briefly introduce the benchmarks and report our detailed experimental settings in the Appendix F.

- **BoardgameQA** is one of the latest benchmarks for logical reasoning. The data is synthesised from formal logical reasoning rules and clauses. The problem provides a self-contained context, pieces of evidence, reasoning rules, and questions. Given the evidence and reasoning rules, LLM must decide if the queried clause is true, false, or unknown. Moreover, BoardgameQA has cases where some evidence might contradict others. To deal with this case, BoardgameQA also provides preferences among the reasoning rules, meaning that the conclusions drawn from the preferred rules have higher priority. In addition, since BoardgameQA does not provide CoT data for unknown questions, we use GPT-4 to generate CoT reasoning trajectories for unknown problems for the training dataset. Thus, the training data is consistent. The prompt for data generation is in the Appendix F.1.
- **MATH** is a challenging mathematical reasoning benchmark, including advanced-level algebra and geometry. In each of the problems, the necessary evidence is provided in the context, and LLM is required to learn correct mathematical theorems and rules to connect the pieces of evidence in the given problem and draw conclusions. It typically requires long reasoning steps in which each mathematical theorem and rule is correctly applied to get the correct answers.
- **Blocksworld** is a symbolic planning problem. It defines a domain with a few blocks in different colours. The problem requires the LLM to rearrange the blocks step-by-step and make the configuration of the blocks satisfy some preferred constraints. Concretely, it requires the LLM to output a sequence of pick and place behaviours, which transform the current configuration of blocks to the goal configuration. Blocksworld has been proved to be NP-hard in finding optimal (i.e., shortest) planning trajectories (Gupta & Nau, 1992), while suboptimal policies might be easier to learn.

**Baselines.** To validate the effectiveness of AUTO-CEI, we conduct extensive experiments comparing it against multiple baselines.

- **SFT.** We perform supervised fine-tuning on the original training dataset $D$ and use the overall accuracy as a reference metric.
- **Vanilla EI.** As noted by Havrilla et al. (2024), Expert Iteration achieves strong results in enhancing reasoning capabilities. We apply Expert Iteration to improve LLM reasoning, evaluating its performance by retaining only correct and assertive answers during the iteration process.
- **SFT + R-Tuning** (Zhang et al., 2023). This is an SFT-based post-training method for hallucination mitigation. Please see our initialisation part in Section 4.1 for details.
- **EI + R-Tuning**. This baseline uses Vanilla Expert Iteration to boost the performance of LLM reasoning first and then uses R-Tuning for post-training.
- **RLKF (DPO)**. Reinforcement Learning from Knowledge Feedback (RLKF) (Xu et al., 2024) is an RL-based post-training method for hallucination mitigation. It aligns the LLM's behaviours according to the consistency and correctness of LLM's sampled responses: it teaches LLM to respond assertively if its responses are correct and consistent and acknowledges "I don't know" if

its responses are mainly wrong or inconsistent. We implemented a DPO version of this method. Please see the Appendix for details.

**Metrics.** We evaluate LLM's responses' overall accuracy, precision, and refusal rate (refusal rate). The overall accuracy measures the overall performance of LLM in a specific reasoning task. A higher accuracy means the LLM can solve more problems. We compute the accuracy by $P_{\text{Acc}} = \frac{\text{Number of correct responses}}{\text{Number of all responses}}$. The precision measures the accuracy of LLM when it is willing to answer the questions assertively. It reflects the overall reliability of LLM reasoning. We compute the precision by $P_{\text{Pre}} = \frac{\text{Number of correct responses}}{\text{Num of all responses} - \text{Num of IDK}}$. In addition, the hallucination (error) rate refers to $\frac{\text{Number of non-refusal but incorrect responses}}{\text{Num of all responses} - \text{Num of IDK}}$. The refusal rate measures how conservative the LLM policy is. It is computed by $P_{\text{IDK}} = \frac{\text{Num of refusal responses}}{\text{Num of all responses}}$. A good LLM policy should have reasonable precision and refusal rate balance, i.e., a high precision and a reasonable refusal rate according to the difficulty of the task.

**Implementation details.** We use Llama-3.1-8B-instruct (Dubey et al., 2024) as the backbone model and use Lora ($r = 128, \alpha = 64$) to fine-tune. The experiments are conducted in a server with $8 \times$ Nvidia A100 (40GB) GPUs. We use DeepSpeed Stage 2 to conduct the training. See Section G in Appendix for more details about the AUTO-CEI setup.

## 5.2 RESULTS

The main result is reported in Table 1. In this result, AUTO-CEI produces high precision (Pre) and keeps a reasonably lower refusal rate (IDK) across all tasks. It also has the highest objective function value $f$ defined by Equation 2 ($\lambda = 0.2$). We summarize our findings in the following paragraphs.

Table 1: Main results. Acc: accuracy (%); Pre: precision (%); IDK: refusal rate (%); $f$: objective function.

| Method | BoardgameQA | | | | MATH | | | | Blocksworld | | | |
|---|---|---|---|---|---|---|---|---|---|---|---|---|
| | Acc | Pre | IDK | $f$ | Acc | Pre | IDK | $f$ | Acc | Pre | IDK | $f$ |
| SFT | 60.23 | 60.23 | – | – | 38.65 | 38.65 | – | – | 51.99 | 51.99 | – | – |
| Vanila EI | 67.77 | 67.77 | – | – | 42.04 | 42.04 | – | – | 71.04 | 71.04 | – | – |
| SFT + R-Tuning | 55.23 | 80.36 | 31.27 | 0.780 | 19.49 | 60.67 | 67.88 | 0.550 | 38.54 | 90.69 | 57.51 | 0.811 |
| EI + R-Tuning | 57.76 | 80.77 | 28.49 | 0.789 | 29.57 | 55.80 | 47.02 | 0.552 | 67.92 | 93.95 | 27.71 | **0.896** |
| RLKF (DPO) | 49.32 | 54.17 | 8.95 | 0.615 | 40.02 | 42.19 | 5.14 | 0.527 | 35.24 | 38.08 | 8.21 | 0.486 |
| AUTO-CEI (Ours) | 59.70 | 84.52 | 29.37 | **0.817** | 35.56 | 55.63 | 36.08 | **0.575** | 74.78 | 91.53 | 18.30 | **0.896** |

**EI improve LLM reasoning.** In our experiments, the Vanila EI produced higher overall accuracy than SFT in all tasks, which indicates the overall capacity of LLM in reasoning is improved. This improvement is because EI helps LLM to sample various trajectories and collect those who draw a correct solution and learn those solutions; for the next iteration of EI, it samples more trajectories near those correct trajectories and keeps collecting and learning. Over time, LLM learns to start from various trajectories and still get to the correct solution, even though some might be suboptimal. Thus it becomes more robust to the randomness in token sampling when generating responses and the resulting compounding errors.

**R-Tuning makes LLM over-conservative.** We found that SFT + R-Tuning and EI + R-Tuning all show the trend of over-conservativeness (i.e., laziness). The results of both baselines have a relatively high refusal rate (IDK) and lower overall accuracy. The low overall accuracy indicates that LLM's capacity has been limited, and LLM produces refusal responses to many problems that lie within the LLM's capability.

**RLKF degrades reasoning performance.** Reported by Xu et al. (2024), RLKF performs well for short, knowledge-grounded arithmetic problems. However, for reasoning problems, the responses generally become much longer for more complex questions. This causes a few additional issues for RLKF techniques. First, the LLM finds it hard to distinguish the correct chosen/rejected responses given the long response length if the dataset is not very large; thus, the reward accuracy is relatively low (it only has 30% accuracy for Blocksworld and 70% for BoardgameQA and MATH). Second, the original strategy of reward shaping greedily maximizes the reward for correct answers and minimises incorrect or refusal responses. As such, the LLM will be over-assertive. As it greedily imitates the

optimal responses guided by the inaccurate reward, it will suffer more from the compounding error when facing the unseen context of the reasoning problems, thus degrading its performance.

**AUTO-CEI improves the LLM reasoning.** The result shows that the overall accuracy of AUTO-CEI is higher than SFT + R-Tuning. Since our method initialises its optimisation using SFT + R-Tuning, this result indicates that AUTO-CEI improves LLM's reasoning capacities. It is because AUTO-CEI gradually adjusts the curriculum to encourage LLM to solve more problems within its capacities and only accept refusal responses after sufficient reasoning attempts. It thus avoids the LLM's laziness in easy problems and improves the robustness of the LLM's problem-solving when it samples suboptimal trajectories. The improvement of precision and refusal rate within a few expert iterations is shown in Figure 7 of the Appendix.

**AUTO-CEI estimates LLM's limit in reasoning and produces the best alignment between its behaviours and its limit.** We compare the error (hallucination) rate and refusal rate (IDK) according to the response length. Figure 3 shows that the error rate of SFT and EI grows exponentially with the increase in response length. On the other hand, we can also see that AUTO-CEI has a relatively uniform low rate of hallucination for different response lengths, and its refusal rate (refusal rate) grows according to the error rate of SFT/EI, as is ideally expected. This result indicates that AUTO-CEI is indeed able to estimate LLM's current capability limit and is further capable of reaching a reasonable alignment between its assertive and conservative behaviours according to its limit. It thus behaves assertively for problems within its limit (low refusal rate for short reasoning length) and conservatively for problems beyond its limit (high refusal rate for long reasoning length). Overall, it produces reliable reasoning while simultaneously maintaining maximum reasoning ability.

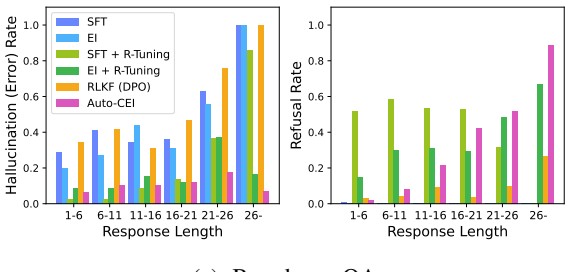

(a). BoardgameQA

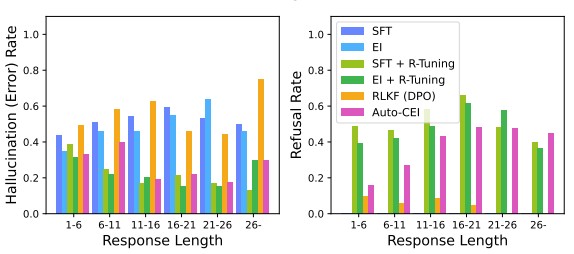

(b). MATH

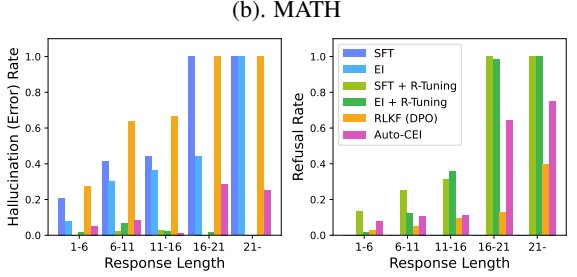

(c). Blocksworld

Figure 3: The error and refusal rate (hallucination/IDK) according to the response length on different datasets.

## 5.3 ABLATION STUDY & DISCUSSION

We conduct an ablation study to verify the effectiveness of our solution design choices and further discuss our underlying assumptions. As the effect of Expert Iterations has been demonstrated in the main result, in this section, we mainly focus on the effect of curriculums and the practical usage of hyper-parameter $\lambda$. The result of the ablation study is shown in Table 2.

Table 2: Ablation study. Acc: accuracy (%); Pre: precision (%); IDK: refusal rate (%).

| Method | BoardgameQA | | | MATH | | | Blocksworld | | |
|---|---|---|---|---|---|---|---|---|---|
| | Acc | Pre | IDK | Acc | Pre | IDK | Acc | Pre | IDK |
| No Curriculum | 56.10 | 85.56 | 34.43 | 27.14 | 52.06 | 47.86 | 68.43 | 93.42 | 26.75 |
| $\lambda = 0.5$ | 59.57 | 81.12 | 26.56 | 37.62 | 49.90 | 24.61 | 79.31 | 84.25 | 5.86 |
| $\lambda = 0.2$ | 59.70 | 84.52 | 29.37 | 35.56 | 55.63 | 36.08 | 74.78 | 91.53 | 18.30 |

**Effect of $\lambda$.** The hyperparameter $\lambda$ decides the optimal objective function and when to stop the optimisation. A higher $\lambda$ will try to minimise the refusal rate (refusal rate) and thus make the LLM more assertive. The lower $\lambda$ will try to focus on maximising precision and tends to keep a higher

refusal rate, making the LLM policy more conservative. The $\lambda$ does not affect the overall training process in AUTO-CEI, and the decision of the hyperparameter is mainly determined by the user's demand. Empirically, $\lambda = 0.2$ has a reasonable balance between the precision and refusal rate. In practice, we suggest the user start by $\lambda = 0.2$ and further adjust the parameter according to the demands. For example, a lower $\lambda$ would be preferred in highly risky tasks, whereas a higher $\lambda$ is better for training search heuristics that require more assertive behaviours, a higher $\lambda$ is better.

**Effect of curriculum.** We aim to use the curriculum to push the limit of LLM reasoning, thus achieving a good alignment of its behaviours and maximizing its reasoning ability simultaneously. We compare the ablation that does not gradually update the reward function defined in Equation 1. The result in Table 2 (No Curriculum) shows that LLM converge to a suboptimal point where its overall accuracy is lower and its refusal rate is higher. In addition, there might be a chance that LLM directly reaches the optimal $f$ after one curriculum with some other choices of $\lambda$. However, as discussed previously, the choice of $\lambda$ is determined by the user's demand, and the optimisation conducted by the curriculum should be able to find the optimal point specified by different $\lambda$.

## 6 CONCLUSION

This paper focused on mitigating LLM's hallucinations and laziness in reasoning tasks. We propose Automatic Curriculum Expert Iteration (AUTO-CEI) to push the limits of LLM's reasoning capacities and align its assertive and conservative behaviours according to these limits for reliable reasoning. We conduct experiments on various challenging reasoning benchmarks, including logical reasoning, mathematics, and planning. Our result suggests that AUTO-CEI maximizes its reasoning ability while achieving a superior alignment between response behaviours and this ability. The fundamental assumption under AUTO-CEI is that solving difficult reasoning problems requires more reasoning steps, thus making the number of reasoning steps a good estimation measure of difficulties. Recent research works by OpenAI (2024) and Snell et al. (2024) suggest that a longer reasoning length in the inference time scales up the LLM reasoning ability and enables it to achieve expert-level capabilities. These researches provide evidence for our assumption. Meanwhile, they also show the chance that our method can also be applied to improve the reliability of those language models with super-long reasoning lengths.

**Limitations.** This work focuses on challenging reasoning problems, where complexity can be estimated by the number of reasoning steps. For simpler problems with shorter reasoning paths or well-studied datasets (e.g., GSM8K), R-Tuning reliably performs well, leaving limited scope for further optimization. This is partly due to the reduced impact of compounding errors in these easier tasks. We also found that the model might decline to answer even if its response is correct. Thus, further adjustment and fine-tuning should be applied to solve this issue, and our method should be able to be extended accordingly as well. In addition, our method might require a long time for optimisation as it requires running Expert Iteration multiple times. Nonetheless, compared to the cost of pre-training, the overhead of the iterative post-training algorithm remains worthwhile for achieving better alignments. Our primary goal was to establish a proof-of-concept using straightforward, reproducible metrics common in curriculum learning. While we focused on the number of reasoning steps as the core metric, the framework can accommodate more sophisticated approaches. Future work could incorporate advanced metrics such as LLM-predicted statistics (Wang et al., 2024b; Damani et al., 2024) for fine-grained curriculum learning. The foundational nature of Auto-CEI makes it amenable to such extensions while maintaining its core benefits of automation and scalability.

**Reproducibility Statement.** In Section 4, we provide pseudocode for our algorithm and explain the details of each part. We provide sufficient details in Appendix Section G for our implementation of AUTO-CEI and training settings. For our experiment benchmarks, we introduce our benchmark selection and metric computation in Section 5.1 and provide the link to download the dataset or the source code from the published papers to generate the dataset, the format of the dataset, and the dataset size we choose in Appendix Section F. We also provide the baseline methods' details, implementation and training in Appendix Section H. Codes are available.

ACKNOWLEDGMENT

We thank all Program chairs, Area chairs, and Reviewers for helpful discussion.

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

## A    DIFFICULTY AND RESPONSE LENGTH

The underlying assumption of our paper is that difficult problems require longer responses to get the answer. We use the MATH dataset, which has manually labelled difficulties, to verify the hypothesis. We measure the steps of reasoning with respect to the labelled difficulty, shown in Fig 4. The result suggests that the reasoning length and manually labelled difficulty are correlated with statistical significance. The scale looks different from Fig 3 in the paper, as the reasoning length distribution is a long tail distribution. The distribution is visualised in Fig 5.

Human-labelled difficulty often diverges from computational measures, with LLM-perceived difficulty better aligning with reasoning steps (Pearson's $r = 0.349$, $p < 1e\text{-}142$). Despite the noise in the MATH dataset's non-standardized steps, Auto-CEI performs robustly, effectively balancing assertiveness and conservativeness.

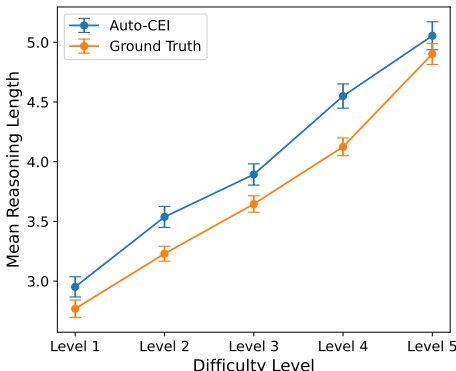

Figure 4: Correlation between reasoning length and labeled difficulty (MATH dataset). For each difficulty level, the mean reasoning length is plotted with error bars representing the standard error of the estimator.

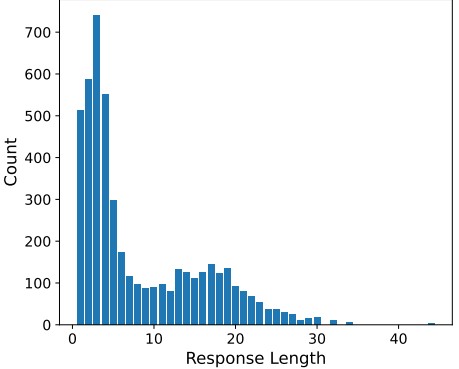

Figure 5: Histogram of reasoning lengths generated by AUTO-CEI for the MATH dataset.

Some difficult problems might have short optimal solutions. However, this doesn't mean the computational complexity of that problem is low. Take NP-complete problems as an example. Searching for the solution may require exponential time (in the worst case), but verifying the solution only takes polynomial time, as the optimal solution is much shorter than the search process. While the question of whether verification is inherently simpler than finding a solution (NP ?= P) remains open, our hypothesis relies on human intuitive common sense: more difficult problems generally require higher computational complexity to arrive at correct solutions, which is also the case for real-world problems and solutions.

## B   HYPERPARAMETER TUNING

The mitigation of hallucination and laziness contradict each other. Optimizing precision makes the LLM respond "I don't know" most of the time; minimising the refusal rate makes the LLM try to make an assertive response all the time. Because of this, our objective is not to maximise accuracy but to achieve the sweet spot of balance between assertiveness and conservativeness.

The hyper-parameter does not affect the training process but determines when to stop the Auto-CEI. Empirically, the user can start an initial hyper-parameter. After it terminates, if the user is unsatisfied with the result, they can further update the hyper-parameter and continue the training process. The previous models can be reused so that no extra computation is involved. One example is shown in Fig 6.

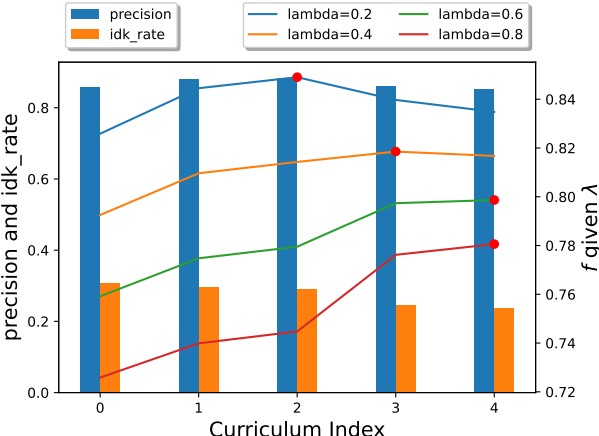

Figure 6: The value of objective function $f$ changes according to the curriculum and hyper-parameter $\lambda$. The red dot denotes the highest value of $f$ in the curve.

## C   REFUSAL ACCURACY

We evaluate if the language model makes a refusal response even if its answer is correct. Thus, we define the refusal accuracy (IDK Acc) below.

$$P_{\text{IDKAcc}} = \frac{\text{Number of refusal response with correct answer}}{\text{Number of refusal response}} \qquad (3)$$

The refusal accuracy measures the proportion of the correct answer in the refusal responses. A lower accuracy indicates a better self-awareness in correctness, while a higher correctness means the language model is over-conservative. The refusal accuracy, compared with the precision, is shown in Table 3.

The result suggests that Auto-CEI's refusal responses have significantly lower accuracy. We also highlight that a model's true capacity is inherently a hidden variable and can only be estimated indirectly, as there is no ground-truth label of which questions are truly beyond LLM's capacity. 1 - (refusal accuracy) is a Monte-Carlo estimation of the model's awareness in providing an inaccurate answer.

Table 3: Refusal accuracy v.s. Precision. Pre: precision (%); IDK Acc: refusal accuracy (%).

|  | Boardgame QA | | MATH | | BlocksWorld | |
| --- | --- | --- | --- | --- | --- | --- |
|  | Pre | IDK Acc | Pre | IDK Acc | Pre | IDK Acc |
| SFT + R-Tuning | 80.36 | 19.62 | 60.67 | 14.27 | 90.69 | 32.44 |
| EI + R-Tuning | 80.77 | 16.67 | 55.80 | 12.74 | 93.95 | 41.21 |
| RLKF | 54.17 | 21.48 | 42.19 | 13.49 | 38.08 | 24.47 |
| Auto-CEI | 84.52 | 12.25 | 55.63 | 9.89 | 91.53 | 27.02 |

## D  GENERALIZABILITY CROSS LANGUAGE MODELS

We conduct additional experiments to test the generalizability of our method in other language models. We did experiments on Mistral-7B-Instruct-v3 on the BoardgameQA benchmark. The result is shown in Table 4. It provides further evidence of our method's generalisability.

Table 4: Additional Experiment on Mistral-7B-Instruct-v3. Acc: accuracy (%); Pre: precision (%); IDK: refusal rate (%); IDK Acc: refusal accuracy (%).

|  | Acc | Pre | IDK | IDK Acc | $f(\lambda = 0.2)$ |
| --- | --- | --- | --- | --- | --- |
| SFT | 71.28 | – | – | – | – |
| EI | 75.54 | – | – | – | – |
| SFT + R-Tuning | 59.02 | 76.39 | 22.74 | 15.42 | 0.7656 |
| EI + R-Tuning | 61.62 | 80.39 | 23.35 | 14.23 | 0.7964 |
| Auto-CEI | 75.01 | 93.47 | 19.75 | 11.50 | 0.9083 |

## E  IMPROVEMENT IN EXPERT ITERATION

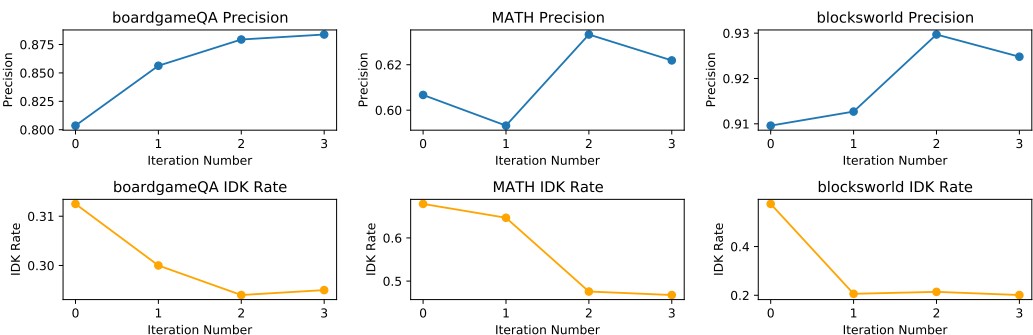

Figure 7: Improvement within EI. The precision and refusal rate converge after 3 iterations.

## F  EXPERIMENTAL SETTINGS

### F.1  BOARDGAMEQA

For boardgameQA, we use the main dataset to do fine-tuning and evaluation [2]. We take the dataset under the folder `BoardgameQA-Main-depth1`, `BoardgameQA-Main-depth2` and `BoardgameQA-Main-depth3`. We show one example of the problem in BoardgameQA in Listing 1. For more details about the dataset itself, please refer to the paper by Kazemi et al. (2024).

In our experiment, we put the `example` as the input prompt and verify if the final answer is the same as the `label`. The `proof` provides the chain of thought inference data, which is used in our fine-tuning.

---

[2]The dataset is downloaded from `https://storage.googleapis.com/gresearch/BoardgameQA/BoardgameQA.zip`

Listing 1: BoardgameQA Problem Example

```
{
    "facts": "The cow learns the basics of resource management from the
aardvark. The dog burns the warehouse of the koala. The dog proceeds to
the spot right after the leopard. The dog reduced her work hours recently
. The hummingbird has a card that is red in color.",
    "rules": "Rule1: For the halibut, if the belief is that the dog shows
 her cards (all of them) to the halibut and the hummingbird knocks down
the fortress of the halibut, then you can add that \"the halibut is not
going to eat the food that belongs to the lion\" to your conclusions.
Rule2: If the hummingbird has a card whose color appears in the flag of
Japan, then the hummingbird knocks down the fortress that belongs to the
halibut. Rule3: If you see that something proceeds to the spot right
after the leopard and burns the warehouse that is in possession of the
koala, what can you certainly conclude? You can conclude that it also
shows her cards (all of them) to the halibut. Rule4: If the cow learns
elementary resource management from the aardvark, then the aardvark burns
 the warehouse of the elephant. Rule5: If at least one animal burns the
warehouse of the elephant, then the halibut eats the food that belongs to
 the lion.",
    "preferences": "Rule5 is preferred over Rule1. ",
    "example": "A few players are playing a boardgame. The current state
of the game is as follows. The cow learns the basics of resource
management from the aardvark. The dog burns the warehouse of the koala.
The dog proceeds to the spot right after the leopard. The dog reduced her
 work hours recently. The hummingbird has a card that is red in color.
And the rules of the game are as follows. Rule1: For the halibut, if the
belief is that the dog shows her cards (all of them) to the halibut and
the hummingbird knocks down the fortress of the halibut, then you can add
 that \"the halibut is not going to eat the food that belongs to the lion
\" to your conclusions. Rule2: If the hummingbird has a card whose color
appears in the flag of Japan, then the hummingbird knocks down the
fortress that belongs to the halibut. Rule3: If you see that something
proceeds to the spot right after the leopard and burns the warehouse that
 is in possession of the koala, what can you certainly conclude? You can
conclude that it also shows her cards (all of them) to the halibut. Rule4
: If the cow learns elementary resource management from the aardvark,
then the aardvark burns the warehouse of the elephant. Rule5: If at least
 one animal burns the warehouse of the elephant, then the halibut eats
the food that belongs to the lion. Rule5 is preferred over Rule1. Based
on the game state and the rules and preferences, does the halibut eat the
 food of the lion?",
    "proof": "We know the cow learns the basics of resource management
from the aardvark, and according to Rule4 \"if the cow learns the basics
of resource management from the aardvark, then the aardvark burns the
warehouse of the elephant\", so we can conclude \"the aardvark burns the
warehouse of the elephant\". We know the aardvark burns the warehouse of
the elephant, and according to Rule5 \"if at least one animal burns the
warehouse of the elephant, then the halibut eats the food of the lion\",
and Rule5 has a higher preference than the conflicting rules (Rule1), so
we can conclude \"the halibut eats the food of the lion\". So the
statement \"the halibut eats the food of the lion\" is proved and the
answer is \"yes\".",
    "goal": "(halibut, eat, lion)",
    "theory": "Facts:\n\t(cow, learn, aardvark)\n\t(dog, burn, koala)\n\t
(dog, proceed, leopard)\n\t(dog, reduced, her work hours recently)\n\t(
hummingbird, has, a card that is red in color)\nRules:\n\tRule1: (dog,
show, halibut)^(hummingbird, knock, halibut) => ~(halibut, eat, lion)\n\
tRule2: (hummingbird, has, a card whose color appears in the flag of
Japan) => (hummingbird, knock, halibut)\n\tRule3: (X, proceed, leopard)^(
X, burn, koala) => (X, show, halibut)\n\tRule4: (cow, learn, aardvark) =>
 (aardvark, burn, elephant)\n\tRule5: exists X (X, burn, elephant) => (
halibut, eat, lion)\nPreferences:\n\tRule5 > Rule1",
    "label": "proved"
```

```
}
```

**BoardgameQA data argumentation**   For the problems whose `label` is `unknown`, the `proof` only provides one simple statement: *The provided information is not enough to prove or disprove the statement "{the query statement}".* This make the responses inconsistent with other responses whose label is `proved` or `disproved`. Thus, we use GPT-4 to generate the chain of thought data for `unknown` cases. The prompt of our data generation is provided in the Listing 2.

Listing 2: Prompt for data generation

```
prompt="""
You are helping to generate the reasoning for a logical reasoning problem
. The problem has no sufficient evidence to conclude if the question is
proved or disproved, thus is not provable, or unknown. You will need to
generate refutation or exploration reasoning steps to show that there is
no evidence to draw a conclusion. Make sure your answer is clear and
concise. Always end your answer by saying So the statement "..." is not
provable and the answer is "unknown".
Example:
Quesiton: A few players are playing a boardgame. The current state of the
 game is as follows. The cheetah needs support from the swordfish. The
crocodile winks at the moose. The dog burns the warehouse of the raven.
The gecko needs support from the aardvark. The kudu learns the basics of
resource management from the aardvark. The leopard eats the food of the
penguin. The leopard holds the same number of points as the lion. The
viperfish winks at the penguin. The zander respects the doctorfish. The
hare does not attack the green fields whose owner is the hummingbird. The
 rabbit does not knock down the fortress of the sea bass. And the rules
of the game are as follows. Rule1: Be careful when something eats the
food that belongs to the penguin and also needs the support of the lion
because in this case it will surely sing a victory song for the whale (
this may or may not be problematic). Rule2: If the gecko needs the
support of the aardvark and the kudu prepares armor for the aardvark,
then the aardvark will not wink at the oscar. Rule3: If the dog burns the
 warehouse that is in possession of the raven, then the raven shows her
cards (all of them) to the hare. Rule4: If you are positive that one of
the animals does not knock down the fortress that belongs to the sun bear
, you can be certain that it will wink at the oscar without a doubt.
Rule5: If the leopard sings a victory song for the whale, then the whale
attacks the green fields of the tilapia. Rule4 is preferred over Rule2.
Based on the game state and the rules and preferences, does the whale
attack the green fields whose owner is the tilapia? This quesion is not
provable. Please generate the reasoning steps. Make sure your answer is
clear and concise.
Answer:
We know the whale's attack on the green fields of the tilapia depends on
whether the leopard sings a victory song for the whale, as stated in Rule
 5.
For the leopard to sing the victory song, it must both eat the food of
the penguin and need the support of the lion, per Rule 1.
However, the game state does not confirm whether the leopard needs the
lion's support, leaving this condition unresolved.
As none of the other rules (Rules 2, 3, or 4) are relevant to the whale's
 attack or the leopard's behavior, and there is insufficient information
to verify the key condition in Rule 1, the conclusion remains unknown.
So the statement "the whale attack the green fields whose owner is the
tilapia" is not provable and the answer is "unknown".
Question: {} This quesion is not provable. Please generate the reasoning
steps. Make sure your answer is clear and concise.
Answer:
""".format(question)
```

When using GPT-4 to generate the data, we choose `temperature`=0.6 and `top_p`=0.95. The model we selected is `gpt-4-turbo`.

### F.2 MATH

We download the MATH dataset via Hugging Face [3]. It automatically divide the dataset into training, validation, and test set. We used all the training data in our experiment and evaluated it using all validation and testing sets to check if the final answer was the same as the labelled data. For more details about the dataset itself, please refer to the paper by Hendrycks et al. (2021). One example format of the dataset is shown in the Listing 3.

Listing 3: Example data point of MATH dataset

```
{
    'level': 'Level 4',
    'type': 'Counting & Probability',
    'prompt': "The reality game show Survivor is played with 16 people
divided into two tribes of 8. In the first episode, two people get
homesick and quit. If every person has an equal chance of being one of
the two quitters, and the probability that one person quits is
independent of the probability that any other person quits, what is the
probability that both people who quit are from the same tribe?\nThink
step by step and use the format of '#### \\{final answer\\}. \n###' to
complete your answer.",
    'answer': '\\frac{7}{15}',
    'completion': 'There are $\\binom{16}{2} = 120$ ways to pick 2 of the
 16 people to quit.\nThere are $\\binom{8}{2} = 28$ ways for them to both
 be from the first tribe, and $\\binom{8}{2} = 28$ ways for them to both
be from the other tribe, for a total of $28 + 28 = 56$ ways for them both
 to be from the same tribe.\nSo the odds that both people who quit are
from the same tribe is $56/120 = \\boxed{\\frac{7}{15}}$.\n#### \\frac
{7}{15}. \n###'
}
```

### F.3 BLOCKSWORLD

The Blocksworld dataset can be generated using the code in the GitHub repository by Valmeekam et al. (2023)[4]. For our case, we generate domains from 4 blocks to 6 blocks, and randomly samples 500 data points for training. We randomly sample 500 data points for validation and testing sets whose optimal solution length (i.e., ground truth plan) is no longer than ten steps. We uniformly sample the tasks according to the ground truth lengths to form the testing set (i.e., 100 two-step tasks, 100 four-step tasks, ..., and 100 ten-step tasks).

The original format of the chain of thought for Blocksworld planning is unnecessarily long, therefore making it difficult to learn a sufficient policy. We modified the format of the chain of thought to make it more concise and informative. The elementary format is shown below: *"{problem_description}. Since {precondition_text}. Thus, we take action {action} … Since {precondition_text}. Thus, we take action {action}. Since the goal is GOAL. The goal conditions are satisfied. ###"*. One example of the dataset format is shown in the Listing 4.

Listing 4: Example of Blocksworld dataset

```
{
    'instance_id': '3',
    'query': 'I am playing with a set of blocks where I need to arrange
the blocks into stacks. Here are the actions I can do\n\nPick up a block\
nUnstack a block from on top of another block\nPut down a block\nStack a
block on top of another block\n\nI have the following restrictions on my
actions:\nI can only pick up or unstack one block at a time.\nI can only
pick up or unstack a block if my hand is empty.\nI can only pick up a
block if the block is on the table and the block is clear. A block is
clear if the block has no other blocks on top of it and if the block is
not picked up.\nI can only unstack a block from on top of another block
if the block I am unstacking was really on top of the other block.\nI can
```

---

[3]Link: https://huggingface.co/datasets/hendrycks/competition_math
[4]Link: https://github.com/karthikv792/LLMs-Planning.

```
 only unstack a block from on top of another block if the block I am
unstacking is clear.\nOnce I pick up or unstack a block, I am holding the
 block.\nI can only put down a block that I am holding.\nI can only stack
 a block on top of another block if I am holding the block being stacked
.\nI can only stack a block on top of another block if the block onto
which I am stacking the block is clear.\nOnce I put down or stack a block
, my hand becomes empty.\nOnce you stack a block on top of a second block
, the second block is no longer clear.\n\nThe plan correctness is defined
 in terms of states resulting from executing the actions in the plan. An
action is executable in a state when all its preconditions hold in that
state. The state resulting from the action execution consists of
everything in the previous state with the addition and deletion of add
and delete effects of the action. Plan correctness is defined as follows:
 if the first action in the plan is applicable in the initial state, i.e
., its preconditions are all present there; and the second action is
applicable in the state resulting from applying the first action to the
initial state, this process continues until the state resulting from the
application of the last action in the last but one state gives rise to
the final state where all the goals are satisfied.\n\n[STATEMENT]\nAs
initial conditions I have that, the red block is clear, the orange block
is clear, the yellow block is clear, the hand is empty, the yellow block
is on top of the blue block, the red block is on the table, the blue
block is on the table and the orange block is on the table\nMy goal is to
 have that the blue block is on top of the orange block and the yellow
block is on top of the red block.\nMy plan is as follows:\n\n[PLAN]',
    'reply': 'Since the yellow block is clear, the hand is empty and the
yellow block is on top of the blue block. Thus, we take action unstack
the yellow block from on top of the blue block. Since the red block is
clear and the hand is currently holding yellow block. Thus, we take
action stack the yellow block on top of the red block. Since the blue
block is clear, the hand is empty and the blue block is on the table.
Thus, we take action pick up the blue block. Since the orange block is
clear and the hand is currently holding blue block. Thus, we take action
stack the blue block on top of the orange block. Since the goal is the
blue block is on top of the orange block and the yellow block is on top
of the red block. The goal conditions are satisfied.\n###',
    'ground_truth_plan': '(unstack d b)\n(stack d a)\n(pick-up b)\n(stack
 b c)\n'
}
```

For more details about the dataset, please refer to the paper by Valmeekam et al. (2023).

## G IMPLEMENTATION DETAILS

### G.1 AUTO-CEI SETTING

**Neural Network Model**   We use Llama-3.1-8B-instruct (Dubey et al., 2024) as the backbone model and use Lora ($r = 128, \alpha = 64$) to fine-tune [5]. We use the Fastchat template designed for Llama-3.1-8B-Instruct. The template is demonstrated in the Listing 5. We fill in the `Prompt` by our query and train the LLM to respond by `Completion`.

Listing 5: Chat Template

```
A chat between a curious human and an artificial intelligence assistant.
The assistant gives helpful, detailed, and polite answers to the human's
questions.
### Human: {Prompt}
### Assistant: {Completion}
###
```

---

[5]The Llama-3.1-8B-instruct model is downloaded at `https://huggingface.co/meta-llama/Llama-3.1-8B-Instruct`.

**Initialisation**   As suggested in the paper, we used R-Tuning (Zhang et al., 2023) as our initialisation strategy. Overall, we use LLM to generate various responses via random sampling. It will produce correct and incorrect responses with intermediate reasoning steps. We collect those incorrect responses and attach an expression of acknowledging limitations. In practice, we use temperature=1.0 and top_p = 0.95 to sample responses. We sample 16 responses in our experiments for each query. The expression of acknowledging limitation is *"Sorry, I am not sure if I answer the question correctly. There might be mistakes in my answer"*.

**Expert Iteration**   In this process, the language model will keep sampling responses, and we select the correct responses or have sufficient reasoning attempts before acknowledging incapability. We use temperature=1.0 and top_p = 0.95 to sample responses and sample 16 responses for each query.

Expert Iteration resamples the response according to the reward function defined in Equation 1. In this function, $c_1$ is initialised by the mean value of reasoning steps produced by the initial LLM policy in the validation set, and $c_2$ is computed to make sure that if the number of reasoning steps has reached $c_1 + 2\sigma$ ($\sigma$ is the standard deviation of the reasoning steps by initial policy), then the reward should be higher than $0.9$. It means the answer is longer than roughly 95% of the reasoning trajectories in the validation set, assuming the distribution of reasoning steps is a normal distribution. We use the keywords to identify the length of the reasoning steps. We set a fixed format for different reasoning tasks, and each task will have a marker for length identification. In general, we use the keyword "Since" to identify the number of reasoning steps for the Blocksworld task, and ".\n" for the BoardgameQA and MATH tasks.

**Curriculum Update**   The curriculum updates the reward function gradually. Specifically, it gradually increases or reduces the value of $c_1$ defined in Equation 1. We use the hill-climbing algorithm to search for the optimal value of the objective function $f$ under the assumption that there is no suboptimal value of $f$ wrt $c_1$. We define the steps of updating $c_1$ by $\min\{0.5, 4\sigma/10\}$, where $\sigma$ denotes the standard deviation of the reasoning length produced by the initial LLM policy. We assume that the domain is $[\mu - 2\sigma, \mu + 2\sigma]$ where $\mu$ denotes the mean length of the initial LLM policy, and the domain determines the step size of updating $c_1$. We also do not want to make the step too large, thus capping it by a threshold value of $0.5$, an empirically selected value. As such, the hill climbing algorithm will search the neighbouring value of $c_1$ according to the step size $d$ and identify if the neighbouring point has a larger $f$ value; otherwise, we believe $c_1$ has reached its optimal point to achieve the highest $f$.

### G.2   TRAINING DETAILS

The experiments are conducted in a server with $8\times$Nvidia A100 GPUs. We use DeepSpeed Stage 2 to conduct the training [6]. During the SFT stage, we choose the batch size = 256, and epoch number is 5 for initial SFT process, and 2 for SFT in R-Tuning and EI process. The learning rate of SFT is $1 \times 10^{-4}$. We use the HuggingFace SFT trainer [7] to conduct the SFT training automatically.

## H   DETAILS OF BASELINE METHODS

### H.1   R-TUNING

The setting is the same as AUTO-CEI in the initialisation part of Section G.1.

### H.2   RLKF

Reinforcement Learning from Knowledge Feedback (RLKF) (Xu et al., 2024) is an RL-based post-training method for hallucination mitigation. It aligns the LLM's behaviours according to the consistency and correctness of LLM's sampled responses: it teaches LLM to respond assertively if its responses are correct and consistent and acknowledges "I don't know" if its responses are mainly wrong or inconsistent. We implemented a DPO version of this method as one of our baseline.

---

[6]The configuration of DeepSpeed Stage 2 can be found at `https://github.com/huggingface/transformers/blob/main/tests/deepspeed/ds_config_zero2.json`

[7]`https://huggingface.co/docs/trl/v0.11.1/en/sft_trainer`

**Data collection** Xu et al. (2024) define the pairwise preference to define the reward function:

$$\text{pair} = \begin{cases} (x, y_c) > (x, y_r), & \text{for} \quad \text{All samples are correct;} \\ (x, y_c) > (x, y_w), & \text{for} \quad \text{There exists correct response in samples;} \\ (x, y_r) > (x, y_w), & \text{for} \quad \text{All samples are wrong.} \end{cases} \tag{4}$$

In this equation, $x$ refers to the query, and $y$ means the response sampled from LLM. $y_c$ denotes the correct response, $y_w$ denotes the wrong responses, and $y_r$ denotes the refusal response (acknowledging "I don't know"). We use this strategy to build up the dataset. Since we do not have a special out-of-domain dataset in our setting, we only build the reliable in-domain preference dataset. We sample the LLM using `temperature`=1.0 and `top_p`=0.95, and sample 16 instances for each query. The template refusal responses in RLKF is the same as the R-Tuning. If all responses are correct, we will annotate all 16 responses using the refusal template to form 16 pairs of preference data. If there are correct and incorrect responses, we sample 16 preference pairs randomly. If all samples are wrong, we use all of the wrong responses to generate the refusal responses and form 16 pairs of preference data.

**Training** We use the TRL DPOTrainer[8] to conduct the training. Similar to the previous setting, we use Lora ($r = 128, \alpha = 64$) for fine-tuning. The experiment is implemented in $8\times$Nvidia A100 GPUs. The batch size of our training is 256, and the learning rate is $1.0 \times 10^{-5}$, and the epoch number is 5.

---

[8]`https://huggingface.co/docs/trl/main/en/dpo_trainer`

