# OpenReview forum: "Automatic Curriculum Expert Iteration for Reliable LLM Reasoning"
_ICLR.cc/2025/Conference — ICLR 2025 Poster_

### Official Review · Reviewer_d7Jy · 2024-11-03

**Soundness:** 3
**Presentation:** 3
**Contribution:** 4
**Rating:** 8
**Confidence:** 4

**Summary:**

This paper proposes Automatic Curriculum Expert Iteration (AUTOCEI) to enhance LLM reasoning and align responses to the model's capabilities–assertively answering within its limits and declining when tasks exceed them, so as to mitigate hallucination and laziness in reasoning tasks. Through experiments on  BoardgameQA, MATH and Blocksworld with Llama-3.1-8B-instruct, The authors demonstrate the effectiveness of AUTO-CEI, achieving superior alignment by effectively balancing assertiveness and conservativeness.

**Strengths:**

* This paper is well-written and presents clear ideas.

* The idea of aligning responses to the model's capabilities–assertively answering within its limits and declining when tasks exceed them is novel.

* The  motivation behind AUTO-CEI is straightforward and strong.

**Weaknesses:**

* AUTO-CEI introduces additianal training overheads. Considering that the process of AUTO-CEI includes EXPERT ITERATION (each iteration needs large amount of resampling), the additional training overhead can not be ignored. I suggest the authors align the sampling cost between different baselines.

* Limited validation across models. The effectiveness of AUTO-CEI is validated only on Llama-3.1-8B-instruct. Further exploration is needed to assess the generalizability to other models.

* The performance is significantly affected by the hyperparameter λ. Table 2 shows that the performance of AUTO-CEI fluctuates greatly under different λ, which can lead to a considerable amount of additional cost during actual usage.

**Questions:**

See weaknesses.

Additionaly,

(1) Have you tested the correlation between the number of reasoning steps and difficulty, as this is a key assumption of AUTO-CEI? If not, I suggest conducting a test experiment on the MATH dataset, considering it has manually labeled difficulty tags.

typos:
line 286 N -> K

---

> ### Author Response · Authors · 2024-11-20
> **Response (1/2)**
>
> Thank you very much for your feedback!
>
> ---
> **W1**
> > AUTO-CEI introduces additianal training overheads. Considering that the process of AUTO-CEI includes EXPERT ITERATION (each iteration needs large amount of resampling), the additional training overhead can not be ignored. I suggest the authors align the sampling cost between different baselines.
>
> The baselines are trained until they converge to a stable result, and the best ones wrt validation are selected for comparison with our method. The baseline (such as EI + R-Tuning) does not require the same computation budget as Auto-CEI to reach its optimal. We observed that vanilla EI converges after 2-3 iterations, as reported in [1]. Thus, extra computation will not help after convergence. In addition, we also tested the case where the R-tuning method was run in multiple iterations. The result shows that repeated iterations of R-tuning make the LLM policy much more conservative. We provide the best result (highest f) in iterations, but further iterations will not be helpful as it causes laziness. We take the best result among the iterations of baselines to make sure the comparison is fair.
>
> [1] Teaching large language models to reason with reinforcement learning.
>
> ---
> **W2**
> > Limited validation across models. The effectiveness of AUTO-CEI is validated only on Llama-3.1-8B-instruct. Further exploration is needed to assess the generalizability to other models.
>
> We conduct additional experiments on Mistral-7B-Instruct-v3 on the BoardgameQA benchmark. The result is attached below. It shows further evidence of the generalisability of our method. Due to the time limit, we may only conduct limited experiments within the time limit of rebuttal. We will complete the experiments and report the results in future revisions.
>
> |                | Acc    | Pre    | IDK    | IDK Acc | $f(\lambda=0.2)$ |
> |----------------|--------|--------|--------|---------|--------|
> | SFT            | 71.28% | -      | -      | -       |-       |
> | EI             | 75.54% | -      | -      | -       |-       |
> | SFT + R-Tuning | 59.02% | 76.39% | 22.74% | 15.42%  | 0.7656 |
> | EI + R-Tuning  | 61.62% | 80.39% | 23.35% | 14.23%  | 0.7964 |
> | Auto-CEI       | 75.01% | 93.47% | 19.75% | 11.50%  | 0.9083 |
>
> In addition, our pilot study using Llama2 for Proofwriter also shows evidence of this generalizability. Please see our reply to Reviewer HEc4 for more details.
>
> ---
> **W3**
> > The performance is significantly affected by the hyperparameter λ. Table 2 shows that the performance of AUTO-CEI fluctuates greatly under different λ, which can lead to a considerable amount of additional cost during actual usage.
>
> The mitigation of hallucination and laziness contradict each other. Optimizing the precision makes the LLM respond "I don't know" most of the time; minimising the refusal rate makes the LLM try to make an assertive response all the time. Because of this, our objective is not to maximise accuracy but to be able to achieve the sweet-spot of balance between assertiveness and conservativeness.
>
> The hyper-parameter does not affect the training process but only determines when to stop the Auto-CEI. Tuning the hyper-parameter does not entail extra training from scratch. We added Figure 6 in the Appendix in the updated manuscript. This figure shows the changes in Precision and Refusal Rate, as well as the objective function $f$, wrt the curriculums in the Auto-CEI training process for the BoardgameQA task. The figure shows that the objective function $f$ has different optimal points given different $\lambda$. Empirically, the user can start a small or large initial hyper-parameter. After it terminates, if the user is not satisfied with the result, they can further update the hyper-parameter and continue the training process. The previous result can be reused so that no extra computation is involved. The hill-climbing algorithm can update the reward in both directions, so user can either choose a large or small initial hyper-parameter.
>
> Even though we provide flexibility to the users to choose hyper-parameter based on the demands, our experiment suggests that $\lambda = 0.2$ works well across the diverse task.

---

> ### Author Response · Authors · 2024-11-20
> **Response (2/2)**
>
> **Q1**
> > Have you tested the correlation between the number of reasoning steps and difficulty, as this is a key assumption of AUTO-CEI? If not, I suggest conducting a test experiment on the MATH dataset, considering it has manually labelled difficulty tags.
>
> We use the MATH dataset, which has manually labelled difficulties, to verify the hypothesis. We measure the steps of reasoning with respect to the labelled difficulty. The result (mean $\pm$ standard error) is shown in the table below and also visualised in Figure 4 in the latest Appendix. Pearson's correlation coefficients are 0.263 for the ground truth (p-value: $ 8.405 \times 10^{-80} $) and 0.196 for *Auto-CEI* (p-value: $ 1.544 \times 10^{-44} $). These results indicate a statistically significant correlation between reasoning length and manually labeled difficulty.
>
> |              | Level 1           | Level 2           | Level 3           | Level 4           | Level 5           |
> |--------------|-------------------|-------------------|-------------------|-------------------|-------------------|
> | Auto-CEI     | 2.952 $\pm$ 0.084 | 3.538 $\pm$ 0.087 | 3.893 $\pm$ 0.088 | 4.550 $\pm$ 0.101 | 5.055 $\pm$ 0.117 |
> | Ground Truth | 2.768 $\pm$ 0.072 | 3.229 $\pm$ 0.062 | 3.645 $\pm$ 0.069 | 4.124 $\pm$ 0.074 | 4.901 $\pm$ 0.088 |
>
> The scale looks different from Figure 3 in the paper, as the reasoning length distribution is a long tail distribution. The distribution is visualised in Figure 5 of the updated Appendix.
>
> ---
> Thank you again for your feedback and suggestions! We hope our responses and revised paper address any remaining concerns and increase your impression and confidence in our work.  Please let us know if you have any further questions.

---

> ### Comment · Reviewer_d7Jy · 2024-11-22
>
> Thanks for your detailed response. Most of my concerns have been addressed except for the issue regarding the correlation between the number of reasoning steps and difficulty, which is a key assumption of AUTO-CEI. The Pearson's correlation coefficients of 0.263 and 0.169 do not indicate a significant correlation between the two variables. Therefore, based on the experimental results in the rebuttal, I believe that difficulty cannot be accurately measured through reasoning steps. I suggest that the authors consider strengthening the measurement of difficulty through other methods, such as directly predicting with SOTA models [1] or predicting through training models [2].
>
> [1] Make Every Penny Count: Difficulty-Adaptive Self-Consistency for Cost-Efficient Reasoning
>
> [2] LEARNING HOW HARD TO THINK: INPUT-ADAPTIVE ALLOCATION OF LM COMPUTATION

---

> ### Author Response · Authors · 2024-11-22
>
> Thank you for your feedback.
>
> Our analysis shows a statistically significant correlation (p $\ll$ 0.05) between reasoning steps and manual difficulty, though variance in the relationship affects the coefficient magnitude. The number of reasoning steps explains approximately 25% of the variance in manual difficulty, indicating its value as a meaningful, albeit partial, predictor.  We also highlight that it is almost impossible to fully explain human labels with a well-defined factor.
>
> While manual difficulty ratings incorporate multiple factors, we note that human-perceived difficulty may NOT fully align with computational complexity measures. We found that the correlation between length and correctness of LLM's response (SFT baseline) is higher than human-labelled difficulty (Pearson's Correlation coefficient=0.3486652319830693, p-value=1.1543789210139188e-142). Correctness is 0 for a wrong answer and 1 for a correct answer. That means LLM-perceived difficulty can be more aligned with the reasoning steps. Thus, for LLMs as algorithmic systems, the number of reasoning steps serves as a practical and reliable complexity indicator, supported by our empirical results.
>
> In addition, we also noticed that the steps in the labelled answers in the MATH dataset are not standardised. In their answers, one step might contain more than one elementary reasoning operation. That means the labelled steps have random noise, causing higher variances and affecting Pearson's coefficient. Even though the dataset is imperfect, our method still demonstrates better performance that trades off assertiveness and conservativeness, showing the robustness of Auto-CEI.
>
> All of these factors introduce variance to the variables, but the correlation is still statistically significant.
>
> We appreciate the suggested alternative difficulty measures and will discuss them in our paper. While these approaches could enhance Auto-CEI in future work, our primary goal was establishing proof-of-concept using straightforward, reproducible metrics common in curriculum learning. This foundational approach should facilitate further research and framework extensions.

---

> ### Comment · Reviewer_d7Jy · 2024-11-22
>
> Thanks for your response. All of my concerns have been fully addressed. Considering the overall novelty and contribution of this paper, I will increase my score from 5 to 8.

---

> > ### Author Response · Authors · 2024-11-22
> >
> > Thank you very much for your comments and increasing your rating! We really appreciate it. We will incorporate our discussion and your suggestions in our revisions.

---

### Official Review · Reviewer_HEc4 · 2024-11-04

**Soundness:** 3
**Presentation:** 3
**Contribution:** 3
**Rating:** 6
**Confidence:** 3

**Summary:**

This paper enhances LLM calibration ability in reasoning tasks by mitigating hallucinations and reducing lazy behaviors. The method uses a curriculum of expert iteration, balancing answering the question and saying 'I don't know' by adapting the reward system according to the difficulty of reasoning tasks. It shows great balancing in assertiveness and conservativeness.

**Strengths:**

1. The paper presents a reasonable motivation that previous approaches to addressing hallucination problems often suffer from overcorrection, leading to overly conservative responses from LLMs on many questions.

2. The paper introduces an innovative reward mechanism that dynamically balances the choice between responding or refusing to answer based on question difficulty. Subsequent experiments demonstrate that this method effectively handles the trade-off between answering and refusing. Figure 3 also illustrates that as the generated text length increases, more questions are declined.

3. The writing is clear, and the figures are well-designed, enhancing the overall readability of the paper.

**Weaknesses:**

1. The paper’s reward uses the length of the generated text to measure the level of the exploration. This limits the method’s generalizability and scalability, as it’s challenging to find a universal text processing metric for different types of text (e.g., code, tables). Furthermore, reward parameters such as c1 and c2​ also vary across datasets, which makes me concerned about the method’s effectiveness in situations where training and testing distributions are not similar.

2. The experimental results show that even the best method in this paper achieves overall accuracy significantly lower than Vanilla EI. While I understand that incorporating refusal may lead to a drop in accuracy, achieving performance closer to Vanilla EI would be more compelling, as the current method shows that adding a refusal option will significantly reduce the accuracy.

**Questions:**

1. The MATH dataset has manually labeled question difficulty; it would be helpful to include a figure illustrating the correlation between generated text length and question difficulty as a validation of this signal.

2. Is the method effective across datasets with different distributions?

3. There are many established metrics for measuring uncertainty with different confidence levels, such as AP used in R-tuning. The current paper only reports results for the binary case, i.e., answering or refusing to answer. I am curious about the performance of this method at different confidence levels.

---

> ### Author Response · Authors · 2024-11-20
> **Response (1/2)**
>
> Thank you very much for your feedback!
>
> ---
> **W1**
> > The paper’s reward uses the length of the generated text to measure the level of the exploration. This limits the method’s generalizability and scalability, as it’s challenging to find a universal text processing metric for different types of text (e.g., code, tables). Furthermore, reward parameters such as c1 and c2 also vary across datasets, which makes me concerned about the method’s effectiveness in situations where training and testing distributions are not similar.
>
> As suggested in the problem formulation, this paper's main scope is natural language reasoning problems, while other text formats, such as code and table, might be out of our scope. Coding might fit in the problem formulation, but its process is a bit different, which requires code execution, online interaction with "world feedback" (i.e., compiler), and debugging. Essentially, any proxy for measuring computational complexity (reasoning step) would be a good metric for our method in general.
>
> The out-of-distribution issue is a major challenge in general. We tested our method in a new dataset (ProofWriter [1]), which contains an out-of-domain and out-of-distribution testing set. ProofWriter is a logical reasoning benchmark similar to BoardGameQA without contradictory evidence.
>
> Our result shows that it still outperforms the baselines.
>
> |               | OOD (Domain) Precision | OOD (Domain) IDK Rate | OOD (Distribution) Precision | OOD (Distribution) IDK Rate |
> |---------------|----------------------|---------------------|---------------------|--------------------|
> | SFT           | 45.92%               | 0                   | 52.14%              | 0                  |
> | EI            | 47.74%               | 0                   | 53.27%              | 0                  |
> | SFT + RTuning | 53.46%               | 29.17%              | 54.21%              | 25.80%             |
> | EI + RTuning  | 54.52%               | 22.45%              | 59.21%              | 29.35%             |
> | Auto-CEI      | 59.32%               | 24.02%              | 60.31%              | 26.10%             |
>
> The original ProofWriter dataset is relatively simple. Thus, we conduct data argumentation that composes multiple small questions to form complex problems, significantly increasing the steps needed to draw a correct conclusion and improving its difficulty. We use the unseen domain and unseen optimal reasoning depth to test the model, and it exhibits good generalisability.
>
> The unseen domain is the new domain that consists of objects and rules that never appear in the training dataset. The out-of-distribution case consists of questions with much longer optimal steps to solve problems (7 - 10 steps, where the training dataset is 1-6 steps), composed of elementary reasoning problems (up to 5 steps to draw a correct conclusion optimally) in the original dataset. However, as Auto-CEI aims to push the limits w.r.t the training dataset, the result might not show LLM policy's full potential for those out-of-distribution questions, but still better than the baselines.
>
> We used the Llama2 model for this experiment as it is a pilot study. We eventually chose BoardGameQA as it is more difficult and is the latest benchmark.
>
> [1] ProofWriter: Generating Implications, Proofs, and Abductive Statements over Natural Language
>
> ---
> **W2**
> > The experimental results show that even the best method in this paper achieves overall accuracy significantly lower than Vanilla EI. While I understand that incorporating refusal may lead to a drop in accuracy, achieving performance closer to Vanilla EI would be more compelling, as the current method shows that adding a refusal option will significantly reduce the accuracy.
>
> Getting higher accuracy or precision while minimising the refusal rate are contradictory goals, and our empirical study suggests that they cannot be achieved simultaneously. Optimising the precision makes the LLM respond "I don't know" most of the time, whereas minimising the refusal rate makes the LLM try to make an assertive response all the time.  Because of this, our objective is not to maximise accuracy but to achieve the sweet spot of balance between assertiveness and conservativeness.

---

> ### Author Response · Authors · 2024-11-20
> **Response (2/2)**
>
> **W3**
> > The MATH dataset has manually labeled question difficulty; it would be helpful to include a figure illustrating the correlation between generated text length and question difficulty as a validation of this signal.
>
> Thank you very much for your suggestion! We measure the steps of reasoning with respect to the labelled difficulty. The result (mean $\pm$ standard error) is shown in the table below and also visualised in Figure 4 in the latest Appendix. Pearson's correlation coefficients are 0.263 for the ground truth (p-value: $ 8.405 \times 10^{-80} $) and 0.196 for *Auto-CEI* (p-value: $ 1.544 \times 10^{-44} $). These results indicate a statistically significant correlation between reasoning length and manually labeled difficulty.
>
> |              | Level 1           | Level 2           | Level 3           | Level 4           | Level 5           |
> |--------------|-------------------|-------------------|-------------------|-------------------|-------------------|
> | Auto-CEI     | 2.952 $\pm$ 0.084 | 3.538 $\pm$ 0.087 | 3.893 $\pm$ 0.088 | 4.550 $\pm$ 0.101 | 5.055 $\pm$ 0.117 |
> | Ground Truth | 2.768 $\pm$ 0.072 | 3.229 $\pm$ 0.062 | 3.645 $\pm$ 0.069 | 4.124 $\pm$ 0.074 | 4.901 $\pm$ 0.088 |
>
> The scale looks different from Figure 3 in the paper, as the reasoning length distribution is a long tail distribution. The distribution is visualised in Figure 5 of the updated Appendix.
>
> ---
> **Q1**
> > Is the method effective across datasets with different distributions?
>
> Yes, it exhibits out-of-distribution generalisation in our additional experiment result. See our reply above.
>
> ---
> **Q2**
> > There are many established metrics for measuring uncertainty with different confidence levels, such as AP used in R-tuning. The current paper only reports results for the binary case, i.e., answering or refusing to answer. I am curious about the performance of this method at different confidence levels.
>
> We test the model's confidence by saying the keyword "Sorry" before the model finalises its response to test its confidence level and measure the AP score across the benchmarks. The higher logProb of "Sorry" indicates lower confidence. Please see the result below.
>
> |              | BoardgameQA | MATH   | Blocksworld |
> |--------------|-------------|--------|-------------|
> | SFT R-Tuning | 0.5238      | 0.2574 | 0.4714      |
> | EI R-Tuning  | 0.5634      | 0.3271 | 0.7581      |
> | RLKF         | 0.4042      | 0.3901 | 0.3248      |
> | Auto-CEI     | 0.6031      | 0.3933 | 0.7914      |
>
> The results show that Auto-CEI has a better and consistent confidence ranking for correct answers. Even though for MATH, Auto-CEI has lower precision than SFT+R-Tuning since R-Tuning makes LLM policy over-conservative, the AP score of Auto-CEI is higher, indicating that the ranking of Auto-CEI is more consistent.
>
> ---
> Thank you again for your feedback and suggestions! We hope our responses and revised paper address any remaining concerns and increase your impression and confidence in our work. Please let us know if you have any further questions.

---

> ### Author Response · Authors · 2024-11-25
> **A Gental Reminder**
>
> Dear Reviewer HEc4,
>
> We truly appreciate your dedicated time and effort in reviewing our work. As the rebuttal period is coming to an end, we’d love to hear back from you to confirm if our response addresses your concerns. Please don’t hesitate to let us know if you have any further questions, and we’d be happy to assist.
>
> Thank you again for your thoughtful feedback!
>
> Best regards,
>
> Authors

---

> > ### Comment · Reviewer_HEc4 · 2024-11-26
> > **Comment**
> >
> > Thank you for the author's response, which addressed most of my concerns. However, while the authors indicated that generating other text formats (e.g., code or JSON) might be beyond the scope of this paper, this limitation still affects the generalizability of the proposed method. For instance, in LLM agent design, JSON-formatted uncertainty evaluation can be crucial. Even for natural language tasks, using newlines as separators may not be suitable for all scenarios. This heuristic design heavily relies on human prior knowledge. Additionally, in the AP comparison, the improvement over RLKF appears a little bit marginal.
> >
> > In general, I acknowledge the innovation and effectiveness of the proposed method and maintain my positive rating.

---

> ### Author Response · Authors · 2024-11-27
> **Further clarification to the remaining concerns**
>
> Thank you very much for your kind reply! We would like to clarify your remaining concerns further.
>
> ---
>
> > However, while the authors indicated that generating other text formats (e.g., code or JSON) might be beyond the scope of this paper, this limitation still affects the generalizability of the proposed method. For instance, in LLM agent design, JSON-formatted uncertainty evaluation can be crucial. Even for natural language tasks, using newlines as separators may not be suitable for all scenarios. This heuristic design heavily relies on human prior knowledge.
>
> **For domain generalisability**
>
> * Our experiments contain logical reasoning, math, and planning, indicating its generalisability across domains.
> * The principle of Auto-CEI that measures LLM-perceived difficulty by reasoning length and makes alignment accordingly should be generalisable. As verified in our study and many other papers (e.g. [1-3]), LLM makes more mistakes with longer responses in various scenarios, either in reasoning or planning and with different formats.
> * LLM agents and code generation are big topics and may contain various potential technical issues. We certainly cannot address the potential issues in all of those domains in one single paper. However, we believe applying the principle of Auto-CEI in those domains would be great future research directions.
>
> [1] Faith and Fate: Limits of Transformers on Compositionality
>
> [2] LLMs Still Can't Plan; Can LRMs? A Preliminary Evaluation of OpenAI's o1 on PlanBench
>
> [3] On the Empirical Complexity of Reasoning and Planning in LLMs
>
> **For format generalisability**
>
> We are NOT claiming that newlines are the "universal" separators. We use newlines as step separators for convenient purposes and acknowledge that they are imperfect measures with random noise. **Even with noise, Auto-CEI still performs better.**
> * For example, we noticed that the MATH dataset has some noise when quantifying the steps by newline separator, as the steps in the labelled answers in the MATH dataset are not standardised. One step might contain more than one elementary reasoning operation in their answers. That means the labelled steps have random noise, causing higher variances. However, even with the noise, our performance is still better than the baselines, striking a better balance between conservativeness and assertiveness. It shows the robustness of Auto-CEI.
> * BoardgameQA and Blocksworld have better-separated steps, and Auto-CEI performs much better in those tasks, showing the effectiveness of measuring difficulty by an accurate estimate of reasoning length.
>
> We believe any suitable measures of computational complexity can be used in Auto-CEI, and the separator should be determined case by case according to the format of different datasets. Any reasonable measures, such as the token length (universal) or number of lines (customised), could be used in Auto-CEI. Our experiment shows that Auto-CEI is robust to the noise of those difficulty measures: even though the measures could be imperfect, Auto-CEI still provides better alignment.
>
> ---
> > Additionally, in the AP comparison, the improvement over RLKF appears a little bit marginal.
>
> First, please note that in the AP score comparison, Auto-CEI outperforms RLKF in BoardgameQA and Blocksworld **by a very large margin.**
>
> In addition, evidence suggests that **analysing and interpreting the AP score ALONE could be misleading when the AP scores are CLOSE**.
> 1. For MATH dataset, the RLKF produces an over-assertive LLM policy with a low refusal rate and low precision. However, since it has a lower refusal rate, its accuracy is slightly higher than Auto-CEI. This means that even though its overall accuracy is higher, it hallucinates more, and thus, its precision is not well aligned with its confidence, making it less reliable.
> 2. For MATH dataset, Auto-CEI's AP score is higher than its overall accuracy and also a bit higher than RLKF's AP score. This means that its precision is better aligned with its confidence. It also explains why it has much higher precision in our main result, indicating it is more reliable.
> 3. As the reviewer suggests us to refer to the R-Tuning paper and test AP scores, we observe that in the **R-Tuning paper** (https://arxiv.org/abs/2311.09677), **R-Tuning has higher precision in all tasks (Reported in Figure 3 in R-Tuning paper. Please note that the accuracy defined in R-Tuning paper is actually the Precision in our paper). However, for the AP score (Reported in Table 1 in R-Tuning paper), R-Tuning only outperforms baselines by a tiny margin or even worse in some rows.**
>
> All evidence suggests that AP scores should be assessed and interpreted together with the main results reported in our manuscript to draw meaningful conclusions, especially when AP scores are close.
>
> ---
> Again, thank you for your kind response and positive attitude toward our paper! We would love to hear back from you and know if the concern has been addressed.

---

### Official Review · Reviewer_4yYx · 2024-11-04

**Soundness:** 3
**Presentation:** 3
**Contribution:** 2
**Rating:** 6
**Confidence:** 5

**Summary:**

This paper proposes an automatic curriculum expert iteration (AUTO-CEI) to enhance LLM reasoning and align responses to the model's capabilities. Enable the model to answer within its limits and decline when tasks exceed them.
Expert iteration can explore the reasoning trajectories near the LLM policy, guiding incorrect paths back on track to reduce compounding errors and improve robustness.
AUTO-CEI uses the length of reasoning steps to measure difficulty and designs an automatic curriculum for expert iteration that rewards correct reasoning.
AUTO-CEI can automatically estimates the boundary of LLMs' reasoning capacity to achieve a reasonable alignment to maximize capacity and control behaviors.

**Strengths:**

1. Using expert iteration is a good idea.
2. The authors consider the trade off between avoiding hallucinations and avoiding laziness, which is important.
3. This paper aims to avoiding reasoning hallucinations and teaching the model to say IDK to reasoning tasks which beyond its ability.

**Weaknesses:**

1. Although considering the trade off between helpfulness and laziness, the paper control this trade off by hyper-parameters, instead of proposing some principles or methods to chose the optimal hyper-parameters.
2. The evaluation metrics in the paper are incomplete; for example, IDK only measures the proportion of times the model outputs "IDK" without assessing the accuracy of those IDK responses.
3. The experiments in the paper may involve unfair comparisons, as AUTO-CEI conducts multiple searches for EI, resulting in significantly more training steps and data compared to other methods.

**Questions:**

Why the process of update R is called curriculum learning? IMO, I think it is more like a search process and curriculum learning is about first learning elementary stuffs and then complex stuffs.

---

> ### Author Response · Authors · 2024-11-20
> **Response**
>
> Thank you very much for your feedback!
>
> ---
> **W1**
> > Although considering the trade off between helpfulness and laziness, the paper control this trade off by hyper-parameters, instead of proposing some principles or methods to chose the optimal hyper-parameters.
>
> The mitigation of hallucination and laziness contradict each other. Optimizing the precision makes the LLM respond "I don't know" most of the time; minimising the refusal rate makes the LLM hallucinate more and try to make an assertive response all the time. Because of this, our objective is not to maximise accuracy but to be able to achieve the sweet-spot of balance between assertiveness and conservativeness.
>
> The hyper-parameter does not affect the training process but only determines when to stop the Auto-CEI. Tuning the hyper-parameter does not entail extra training from scratch. We added Figure 6 in the Appendix in the updated manuscript. This figure shows the changes in Precision and Refusal Rate, as well as the objective function $f$, wrt the curriculums in the Auto-CEI training process for the BoardgameQA task. The figure shows that the objective function $f$ has different optimal points given different $\lambda$. Empirically, the user can start a small or large initial hyper-parameter. After it terminates, if the user is not satisfied with the result, they can further update the hyper-parameter and continue the training process. The previous result can be reused so that no extra computation is involved. The hill-climbing algorithm can update the reward in both directions, so user can either choose a large or small initial hyper-parameter.
>
> Even though we provide flexibility to the users to choose hyper-parameter based on the demands, our experiment suggests that $\lambda = 0.2$ works well across diverse tasks.
>
> ---
> **W2**
> > The evaluation metrics in the paper are incomplete; for example, IDK only measures the proportion of times the model outputs "IDK" without assessing the accuracy of those IDK responses.
>
> We report the accuracy of the refusal responses here. The result suggests that Auto-CEI's refusal responses have significantly lower accuracy. We will put the new metric into the manuscript in future revisions.
>
> |                | BQA Pre | BQA IDK Acc | MATH Pre | MATH IDK Acc  | BW Pre | BW IDK Acc |
> |----------------|-----------------|---------------------|----------|--------------|-----------------|---------------------|
> | SFT + R-Tuning | 80.36%          | 19.62%              | 60.67%   | 14.27%       | 90.69%          | 32.44%              |
> | EI + R-Tuning  | 80.77%          | 16.67%              | 55.80%   | 12.74%       | 93.95%          | 41.21%              |
> | RLKF           | 54.17%          | 21.48%              | 42.19%   | 13.49%       | 38.08%          | 24.47%              |
> | Auto-CEI       | 84.52%          | 12.25%              | 55.63%   | 9.89%        | 91.53%          | 27.02%              |
>
> **IDK Acc -- False Positive (IDK for correct ones)**
>
> ---
> **W3**
> > The experiments in the paper may involve unfair comparisons, as AUTO-CEI conducts multiple searches for EI, resulting in significantly more training steps and data compared to other methods.
>
> The baselines are trained until they converge to a stable result, and the best ones wrt validation are selected for comparison with our method. The baseline (EI + R-Tuning) does not require the same computation budget as Auto-CEI to reach its optimal. We observed that vanilla EI converges after 2-3 iterations, as reported in [1]. Thus, extra computation will not help after convergence. In addition, we also tested the case where the R-tuning method was run in multiple iterations. The result shows that repeated iterations of R-tuning significantly increase the refusal rate. We provide the best result (highest f) in iterations, but further iterations will not be helpful as it causes laziness. We take the best result among the iterations of baselines to make sure the comparison is fair.
>
> [1] Teaching large language models to reason with reinforcement learning.
>
> ---
> **Q1**
> > Why the process of update R is called curriculum learning?
>
> The spirit of Curriculum Reinforcement Learning is that learning to solve simple tasks can be transferred to solving complex tasks. We achieve the same spirit by manipulating the reward function: we train the LLM in a curriculum that encourages it to solve simple problems (problems that require short reasoning) first and be conservative about complex problems. The reward function controls this process. After it learns to answer simple problems with various reasoning trajectories and gets incorrect reasoning back on track, we gradually transfer its capacity to more complex problems. Thus, we call it Curriculum Expert Iteration.
>
> ---
> Thank you again for your feedback and suggestions! We hope our responses address any remaining concerns and increase your impression and confidence in our work. Please let us know if you have further questions.

---

> ### Author Response · Authors · 2024-11-25
> **A Gental Reminder**
>
> Dear Reviewer 4yYx,
>
> We truly appreciate your dedicated time and effort in reviewing our work. As the rebuttal period is coming to an end, we’d love to hear back from you to confirm if our response addresses your concerns. Please don’t hesitate to let us know if you have any further questions, we’d be happy to assist.
>
> Thank you again for your thoughtful feedback!
>
> Best regards,
>
> Authors

---

> > ### Comment · Reviewer_4yYx · 2024-11-26
> >
> > Thanks for your responses.
> > The rebuttal addressed my concerns about hyper-parameters.
> > Although the low IDK accuracy is still a weakness, I think this paper's quality and contributions are good.
> > Therefore, I will maintain my positive rating.

---

> ### Author Response · Authors · 2024-11-26
>
> Dear Reviewer 4yYx,
>
> Thank you for your feedback! We apologize for any confusion regarding the evaluation of IDK accuracy and have revised our response for clarity. Our IDK accuracy calculate the rate where LLM’s answer is correct but LLM eventually response IDK at the end.
>
> Our motivation is to ensure the model responds with "IDK" only when the question is genuinely beyond its capacity. In this context, **IDK accuracy** serves as a metric analogous to **false positives**. **Lower IDK accuracy indicates better calibration**, which aligns with our goal.
>
> Therefore, achieving low IDK accuracy is a key strength of our approach. We hope this explanation resolves your concerns. Please feel free to reach out with any further questions or feedback.
>
> Best regards,
>
> Authors

---

> ### Comment · Reviewer_4yYx · 2024-11-26
>
> Thanks for the reply.
> Actually I mean there should be a metric to measure the model whether refuse to answer the questions they truly don't know.
> Check the accuracy of normal response and check whether its acc is low is an indirect way.
> But it's also make sense in some way I think.

---

> ### Author Response · Authors · 2024-11-26
>
> Thank you for your swift and helpful response.
>
> Indeed, IDK accuracy is not a direct measure of "the questions the model truly doesn’t know." However, **it remains the most straightforward metric that can be derived from the model's responses.**
>
> We understand that a model's true capacity is inherently a hidden variable and can only be estimated indirectly, as there is **no ground-truth label** of which questions are truly beyond LLM's capacity. **1 - (IDK Accuracy) is a Monte-Carlo estimation of the model's awareness in providing an inaccurate answer**.
>
> If you have any suggestions for improving this estimation, we would greatly appreciate your input. Thanks very much!

---

### Official Review · Reviewer_BWMo · 2024-11-04

**Soundness:** 3
**Presentation:** 4
**Contribution:** 3
**Rating:** 8
**Confidence:** 4

**Summary:**

The paper addresses the problem of hallucinations in reasoning for large language models. Specifically, in methods that use expert iteration for improving this reasoning in LLMs. The authors propose adding a Refusal option to the EI pipeline and rewarding the refusal when the problem has a certain level of difficulty (length of reasoning is used as a proxy for difficulty). Based on the reward that a response gets, the data for the next iteration of EI is selected accordingly. To balance refusal and improvements in reasoning, the authors propose using a curriculum to balance the two objectives. Based on measures of accuracy and refusal rate, the authors show the effectiveness of their method compared to baselines and ablations.

**Strengths:**

- The paper is clearly motivated, the problem of hallucinations in LLM reasoning has not been explored as much.
- The paper presents the problem, the methods and the baselines clearly.
- The results on the three datasets, MATH, blocksworld and boardgameQA are comprehensive and provide sufficient evidence of the utility of the method.
- The authors also present sufficient ablations of their method, with varying versions of the curriculum used.

**Weaknesses:**

- Missing citations for central EI works: STaR (https://arxiv.org/abs/2203.14465), Training Chain-of-Thought via Latent-Variable Inference (https://arxiv.org/abs/2403.04642)
- What other measures of difficulty are possible (very recent paper that the authors can choose to ignore since it came out after the paper deadline: https://arxiv.org/abs/2410.04707)? Can a linear probe decode difficulty? A discussion is needed.
- Similarly, are there other mechanisms of knowing when to say I dont know possible? Is a linear probe enough for refusal? Like in Language models mostly know what they know (https://arxiv.org/abs/2207.05221).
- Where could the current framework of length based difficulty fail? When sampling multiple times, what is the variation in the number of steps needed to reach the answer for a problem?
- Are the number of iterations for Auto-CEI and EI etc matched? More generally, how does the amount of training data / compute required vary across the method and baselines? I think some of the implementation details could be moved to main in the revision. This would help readability.
- How could this method be extended beyond expert iteration, to more online methods of learning like PPO or REINFORCE?

**Questions:**

See weaknesses.

---

> ### Author Response · Authors · 2024-11-20
> **Response (1/2)**
>
> Thank you very much for your feedback!
>
> ---
> **W1**
> > Missing citations
>
> Thanks for your reference. We have updated them to the latest version.
>
> ---
> **W2**
> > What other measures of difficulty are possible (very recent paper that the authors can choose to ignore since it came out after the paper deadline: https://arxiv.org/abs/2410.04707)? Can a linear probe decode difficulty? A discussion is needed.
>
> In our paper, we define the difficulty of a reasoning problem according to its computational complexity. However, similar types of problems can vary significantly in difficulty. For example, the Subset Sum has been proven NP-complete, but not all instances of the Subset Sum need exponential computation. Thus, it is still unclear whether learning to predict the general computational difficulty of a problem is doable, as we cannot assert a clear pattern exists. Our method suggests that the LLM attempt to solve the problem first, thus having a better estimate of the difficulty and making a response accordingly.
>
> The paper suggested by the reviewer, even though it was released after the paper deadline, has a similar definition to ours. It measures the difficulty by the expected improvement if an extra computation budget is given, and more difficult problems require more computation to get better results. Essentially, it still measures the computational complexity of a problem while changing its format for better learning.
>
> ---
> **W3**
> > Similarly, are there other mechanisms of knowing when to say I dont know possible? Is a linear probe enough for refusal? Like in Language models mostly know what they know (https://arxiv.org/abs/2207.05221).
>
> The paper suggested by the reviewer mainly talks about knowledge-grounded question-answering cases. It tests benchmarks that ask factual questions (MMLU) to test LLM's internal knowledge (e.g., What is George W. Bush's father's name?). However, this is NOT the focus of our paper. We focus on the reasoning **hallucination in reasoning steps** without factual hallucination in the evidence. More concretely, reasoning hallucination is about the **incorrect application of the evidence and reasoning rules**, such as the precondition of the reasoning rule not being satisfied by the current evidence or the conclusion drawn from the rule and evidence being wrong.
>
> ---
> **W4**
> > Where could the current framework of length based difficulty fail? When sampling multiple times, what is the variation in the number of steps needed to reach the answer for a problem?
>
> We use the MATH dataset, which has manually labelled difficulties, to verify the hypothesis. We measure the steps of reasoning with respect to the labelled difficulty. The result (mean $\pm$ standard error) is shown in the table below and also visualised in Figure 4 in the latest Appendix. Pearson's correlation coefficients are 0.263 for the ground truth (p-value: $ 8.405 \times 10^{-80} $) and 0.196 for *Auto-CEI* (p-value: $ 1.544 \times 10^{-44} $). These results indicate a statistically significant correlation between reasoning length and manually labeled difficulty.
>
> |              | Level 1           | Level 2           | Level 3           | Level 4           | Level 5           |
> |--------------|-------------------|-------------------|-------------------|-------------------|-------------------|
> | Auto-CEI     | 2.952 $\pm$ 0.084 | 3.538 $\pm$ 0.087 | 3.893 $\pm$ 0.088 | 4.550 $\pm$ 0.101 | 5.055 $\pm$ 0.117 |
> | Ground Truth | 2.768 $\pm$ 0.072 | 3.229 $\pm$ 0.062 | 3.645 $\pm$ 0.069 | 4.124 $\pm$ 0.074 | 4.901 $\pm$ 0.088 |
>
> Some difficult problems might have short optimal solutions. However, this doesn't mean the computational complexity of that problem is low. Take NP-complete problems as an example. Searching for the solution may require exponential time (in the worst case), but verifying the solution only takes polynomial time, as the optimal solution is much shorter than the search process. While the question of whether verification is inherently simpler than finding a solution (NP ?= P) remains open, our hypothesis relies on human intuitive common sense: more difficult problems generally require higher computational complexity to arrive at correct solutions, which is also the case for real-world problems and solutions.

---

> ### Author Response · Authors · 2024-11-20
> **Response (2/2)**
>
> **W5**
> > Are the number of iterations for Auto-CEI and EI etc matched? More generally, how does the amount of training data / compute required vary across the method and baselines?
>
> All baselines and our method use the same-sized training dataset.
>
> The baselines are trained until they converge to a stable result, and the best ones wrt validation are selected for comparison with our method. The baseline (such as EI + R-Tuning) does not require the same computation budget as Auto-CEI to reach its optimal. We observed that vanilla EI converges after 2-3 iterations, as reported in [1]. Thus, extra computation will not help after convergence. In addition, we also tested the case where the R-tuning method was run in multiple iterations. The result shows that repeated iterations of R-tuning make the LLM policy much more conservative. We provide the best result (highest f) in iterations, but further iterations will not be helpful as it causes laziness. We take the best result among the iterations of baselines to make sure the comparison is fair.
>
> ---
> **W6**
> > How could this method be extended beyond expert iteration, to more online methods of learning like PPO or REINFORCE?
>
> It should mostly be applied to other reinforcement learning algorithms. The reward function should remain the same. However, the PPO learns an auxiliary value function, which might not be easily transferred during curriculum learning as the reward function will be updated in the curriculum. Expert iteration is a good choice as it is simple to implement and faster to train, and it produces similar or even better performance than PPO in reasoning tasks [1].
>
> [1] Teaching large language models to reason with reinforcement learning.
>
> ---
> Thank you again for your feedback and suggestions! We hope our responses and revised paper address any remaining concerns and increase your impression and confidence in our work. Please let us know if you have any further questions.

---

> ### Author Response · Authors · 2024-11-25
> **A gental reminder**
>
> Dear Reviewer BWMo,
>
> We truly appreciate your dedicated time and effort in reviewing our work. As the rebuttal period is coming to an end, we’d love to hear back from you to confirm if our response addresses your concerns. Please don’t hesitate to let us know if you have any further questions, and we’d be happy to assist.
>
> Thank you again for your thoughtful feedback!
>
> Best regards,
>
> Authors

---

> > ### Comment · Reviewer_BWMo · 2024-11-25
> > **Thank you for your response**
> >
> > Thank you for your response!! The results with difficulty are really interesting.
> >
> > I will maintain my initial positive evaluation.

---

### Author Response · Authors · 2024-11-20
**Overall response**

We thank all the reviewers' efforts in reading our paper and providing valuable feedback! In this message, we summarised the key concerns and suggestions from the reviewers and our responses.

---
**1. Hyperparameters**

Reviewer 4yYx and d7Jy are concerned about the hyper-parameter tuning.

The mitigation of hallucination and laziness contradict each other. Optimizing precision makes the LLM respond "I don't know" most of the time; minimising the refusal rate makes the LLM try to make an assertive response all the time. Because of this, our objective is not to maximise accuracy but to achieve the sweet spot of balance between assertiveness and conservativeness.

The hyper-parameter does not affect the training process but determines when to stop the Auto-CEI. Empirically, the user can start an initial hyper-parameter. After it terminates, if the user is unsatisfied with the result, they can further update the hyper-parameter and continue the training process. The previous models can be reused so that no extra computation is involved.

See our reply to Reviewer 4yYx and d7Jy for more details.

---
**2. Fair comparison in computation budget**

Reviewers 4yYx and d7Jy point out the computational overhead entailed by our method and are concerned about the unfair comparison.

The baselines are trained until they converge to a stable result, and the best ones wrt validation are selected for comparison with our method. The baseline (such as EI + R-Tuning) does not require the same computation budget as Auto-CEI to reach its optimal. We observed that vanilla EI converges after 2-3 iterations, as reported in [1]. Thus, extra computation will not help after convergence. In addition, we also tested the case where the R-tuning method was run in multiple iterations. We take the best result among the iterations of baselines to make sure the comparison is fair.  See our reply to 4yYx and d7Jy for more details.

[1] Teaching large language models to reason with reinforcement learning.

---
**3. Correlation between difficulty and reasoning steps**

Reviewer BWMo, HEc4 and d7Jy suggest further verifying the correlation between difficulty and reasoning length.

We use the MATH dataset, which has manually labelled difficulties, to verify the hypothesis. We measure the steps of reasoning with respect to the labelled difficulty. The result suggests that the reasoning length and manually labelled difficulty are correlated with statistical significance. See our reply to BWMo, HEc4 and d7Jy for more details.

UPDATE:
Human-labelled difficulty often diverges from computational measures, with LLM-perceived difficulty better aligning with reasoning steps (Pearson's r = 0.349, p < 1e-142). Despite the noise in the MATH dataset's non-standardized steps, Auto-CEI performs robustly, effectively balancing assertiveness and conservativeness. See our reply to d7Jy for more details.

---
**4. Results across LLMs**

Reviewer d7Jy wishes to verify Auto-CEI's generalisability across different LLM models.

We conducted additional experiments on Mistral-7B-Instruct-v3 on the BoardgameQA benchmark. The result shows further evidence of our method's generalisability. Due to the time limit, we may only conduct part of the experiments during rebuttal. We will complete the experiments and report the results in future revisions. Our results (demonstrated below) using Llama2 for Proofwriter also show evidence of this generalizability. See our reply to d7Jy for more details.

---
**5. Out-of-distribution and out-of-domain generalization**

Reviewer HEc4 wish to see the out-of-distribution performance.

We tested our method in a new logical reasoning dataset (ProofWriter [2]) containing an out-of-domain testing set. Our result shows that it still outperforms the baselines. We used the Llama2 model for this experiment, as it is a pilot study. We eventually chose BoardGameQA, which is more difficult and the latest benchmark. See our reply to HEc4 for more details.

[2] ProofWriter: Generating Implications, Proofs, and Abductive Statements over Natural Language

---
Thank you again for your valuable feedback! We hope our responses clarify your concerns and increase your impression and confidence in our work. Please let us know if you have further concerns. We look forward to hearing back from you soon!

---

### Comment · Area_Chair_aS1X · 2024-11-25

Dear Reviewers,

The rebuttal discussion period is coming to a close and the paper currently has a mix of positive and negative reviewers. The authors have spent a lot of time responding to each concern -- can you take a look at the author responses and let them know any remaining concerns you have?

Best,
AC

---

### Meta-Review · Area_Chair_aS1X · 2024-12-23

**Metareview:**

The paper proposes Automatic Curriculum Expert Iteration (AUTO-CEI) to enhance LLM reasoning and align responses to the model's capabilities—assertively answering within its limits and declining when tasks exceed them, so as to mitigate hallucination and laziness in reasoning tasks. Specifically, the work adds a Refusal option (saying “I don’t know“) to the expert iteration pipeline and rewarding the refusal when the problem has a certain level of difficulty (length of reasoning is used as a proxy for difficulty). The authors proposed to use a curriculum to balance between refusal and answering the question. The paper presents a clear motivation (reducing hallucination, improving reasoning), and describes their methods clearly. The approach is sound and the empirical results on MATH, blocks-world and boardgameQA are comprehensive and provide sufficient evidence of the utility of the method.

**Additional Comments On Reviewer Discussion:**

The reviewers are generally positive. The authors managed to address each of the reviewers' questions/concerns properly.

---

### Decision · Program_Chairs · 2025-01-22

Accept (Poster)